# WavePolyp: Video Polyp Segmentation via Hierarchical Wavelet-Based Feature Aggregation and Inter-Frame Divergence Perception

**Yuhua Zhang[1], Guilian Chen[1], Yuanqin He[1], Huisi Wu[1]\*, Jing Qin[2]**

[1]College of Computer Science and Software Engineering, Shenzhen University
[2]Centre for Smart Health, School of Nursing, The Hong Kong Polytechnic University
`2400101085@mails.szu.edu.cn, hswu@szu.edu.cn`

## Abstract

Automatic polyp segmentation from colonoscopy videos is a crucial technique that assists clinicians in improving the accuracy and efficiency of diagnosis, preventing polyps from developing into cancer. However, video polyp segmentation (VPS) is a challenging task due to (1) the significant inter-frame divergence in videos, (2) the high camouflage of polyps in normal colon structures and (3) the clinical requirement of real-time performance. In this paper, we propose a novel segmentation network, *WavePolyp*, which consists of two innovative components: a hierarchical wavelet-based feature aggregation (HWFA) module and inter-frame divergence perception (IDP) blocks. Specifically, HWFA excavates and amplifies discriminative information from high-frequency and low-frequency features decomposed by wavelet transform, hierarchically aggregating them into refined spatial representations within each frame. This module enhances the representation capability of intra-frame spatial features, effectively addressing the high camouflage of polyps in normal colon structures. Furthermore, IDP perceives and captures inter-frame polyp divergence through a temporal divergence perception mechanism, enabling accurate polyp tracking while mitigating temporal inconsistencies caused by the significant inter-frame variations across frames. Extensive experiments conducted on the SUN-SEG and CVC-612 datasets demonstrate that our method outperforms other state-of-the-art methods. Code is available at `https://github.com/FishballZhang/WavePolyp`.

## 1 Introduction

Colorectal cancer (CRC), the third most common cancer worldwide, may have a survival rate exceeding 90% in early stages but drops below 5% in advanced stages (Ji et al., 2022). Colonoscopy is widely used in clinical practice to detect precancerous polyps, which are precursors to CRC, and remove them, and thus improve CRC survival rate (Ahn et al., 2012). However, manual screening of polyps in colonoscopy is labor-intensive, time-consuming, and error-prone (Li et al., 2025), leading to a high polyp miss rate in diagnosis. Therefore, it is highly demanded to develop automatic polyp segmentation methods to improve the accuracy and efficiency of diagnosis.

However, automatically segmenting polyps from colonoscopy videos (also called VPS) is a very challenging task for the following reasons. First, significant variations in polyp size, location, and shape among consecutive frames pose great challenges to track and identify polyps, as shown in fig. 1 (c-e). Second, polyps often originate from the surrounding diseased tissues, leading to a high inherent similarity with the background (also regarded as the high camouflage of polyps), which impedes precise delineation of polyps, as shown in fig. 1 (a-b). Third, it is crucial for VPS to achieve real-time performance to provide timely assistance to physicians during colonoscopy examination.

Recently, many deep learning based approaches have been proposed for image polyp segmentation (IPS). Most IPS methods either leveraged the local feature extraction capabilities of convolutional

---

\*Corresponding Author

Figure 1: Challenges in VPS: (a) and (b) the high camouflage of polyps, (c) position variation, (d) shape variation, (e) size variation. Note that (a)-(e) represent two strictly adjacent frames.

neural networks (CNN) (Wang et al., 2022a; Zhang et al., 2020; Zhong et al., 2020) or harnessed the global context modeling strengths of transformers (Dong et al., 2023; He et al., 2023; Zhang et al., 2021) to segment polyps within static frames. However, these methods demonstrate shortcomings when applied to colonoscopy videos, as they are incapable of learning the temporal dependencies from video sequences. To improve the performance of VPS, it is crucial to model temporal consistency between consecutive frames to improve segmentation performance across frames. In this regard, some VPS approaches propose to integrate hybrid 2D/3D CNN architectures (Puyal et al., 2020) or employ normalized attention mechanisms (Ji et al., 2022; 2021) to capture temporal dependencies. On the other hand, some studies (Hu et al., 2024) leveraged long-term and short-term receptive fields to improve the reliability and stability of spatial-temporal features. However, these methods primarily enhance the intra-frame spatial representations by incorporating inter-frame temporal information, largely ignoring the exploration of discriminative features within individual frames and the perception of inter-frame divergence. As a result, they struggle to comprehensively address VPS challenges due to lack of discriminative features for learning frame-wise polyp details and inter-frame divergent information to recognize dynamic changes across video frames.

In this paper, we propose a novel VPS model, aiming at excavating intra-frame discriminative features from the frequency domain and enhancing the perception of inter-frame divergence; we call it *WavePolyp*. Our model has two innovative components: a hierarchical wavelet-based feature aggregation (HWFA) module and inter-frame divergence perception (IDP) blocks. In HWFA, multilevel features are decomposed into high-frequency (HF) and low-frequency (LF) components using wavelet transform. These frequency components are then enhanced through a dedicated HF and LF calculation unit, which refines their representations to improve feature discrimination. Subsequently, an ascending frequency-guided aggregation unit hierarchically integrates the enhanced frequency cues across levels, enabling *WavePolyp* to effectively capture fine-grained spatial details within individual frames. In IDP, we design a temporal divergence perception mechanism to improve the tracking of polyp targets among consecutive frames, mitigating the interference caused by inter-frame temporal inconsistencies. We conduct extensive experiments on two benchmark datasets: SUN-SEG (Ji et al., 2022) and CVC-612 (Bernal et al., 2015), providing comprehensive comparisons with state-of-the-art (SOTA) methods and demonstrating the effectiveness of our *WavePolyp*. Our major contributions are summarized as follows:

- We propose a novel network for VPS by focusing on the excavation of intra-frame discriminative features and the perception of inter-frame divergence.

- We propose a HWFA module to enhance spatial representations within individual colonoscopy frames based on frequency domain. Additionally, we employ IDP blocks to temporally perceive inter-frame divergence, enabling accurate polyp tracking in colonoscopy videos.

- Our method significantly outperforms other SOTA methods on two polyp video datasets: SUN-SEG and CVC-612, achieving better balance between segmentation accuracy and real-time efficiency.

## 2 RELATED WORK

### 2.1 POLYP SEGMENTATION

With advancements in deep learning, various IPS have been introduced to detect the pixel-level polyp regions from colonoscopy images. Methods (Akbari et al., 2018) based on fully convolution net-

works (FCN) and those (Zhou et al., 2020; Zhang et al., 2020) based on U-Net (Ronneberger et al., 2015) have been proposed to extract precise semantic information. Certain methods (Fang et al., 2019; Wang et al., 2022a) incorporate boundary constraints to achieve boundary-aware segmentation results. Furthermore, in order to enhance global information perception capabilities, researchers (He et al., 2023; Wu et al., 2024) combine transformer (Vaswani et al., 2017) and CNN to simultaneously model global and local information, thereby attaining superior performance. SAMAug (Zhang et al., 2023) leverages SAM (Kirillov et al., 2023) to augment polyp images without fine-tuning its parameters, Polyp-SAM (Li et al., 2024) instead retrains SAM directly on polyp-specific datasets to improve segmentation performance.

However, these IPS methods overlook the temporal cues between adjacent frames in colonoscopy videos. Therefore, some researchers have devoted to integrating spatial-temporal features across consecutive frames. For example, a hybrid 2/3D CNN framework (Puyal et al., 2020) was introduced to account for the aggregation of spatial-temporal correlation. PNS+ (Ji et al., 2022) implemented a global-to-local learning strategy and a normalized self-attention module to model and balance short-term and long-term dependencies. SALI (Hu et al., 2024) leveraged both long-term and short-term receptive fields to model temporal coherence to improve the stability and reliability of spatial-temporal features. VP-SAM (Fang et al., 2024) enhances SAM with a parallel spatio-temporal side network that explicitly models inter-frame motion dynamics using deformable convolutions and cross-attention, thereby improving polyp tracking across video frames. Diff-VPS (Lu et al., 2024) integrated multi-task supervision into diffusion models and incorporated a temporal module using a generative adversarial self-supervised strategy to mitigate the high camouflage of polyps in videos. However, these methods fail to capture the fine-grained intra-frame discriminative feature of highly camouflaged polyps and the divergent information between adjacent frames.

## 2.2 WAVELET TRANSFORM IN DEEP LEARNING

Wavelet Transform (WT), which is essential for signal analysis, has been gradually applied in computer vision tasks. SDWNet (Zou et al., 2021), which utilizes the frequency domain information generated by WT as a complement to the spatial domain information, is applied to the image deblurring task. LAAT (Li et al., 2023) employs a wavelet fusion module to combine shallow structures and deep details to recover realistic images in the frequency domain for face super-resolution. WaveDiff (Phung et al., 2023) uses WT in generative models to enhance the visual quality of generated images as well as to improve computational performance. FEDER (He et al., 2023) addresses the intrinsic similarity of the foreground and background by decomposing the features into different frequency bands using learnable wavelets. Different from the above methods, we apply discrete wavelet transform to decompose multi-level features into HF and LF components. The decomposed features are hierarchically enhanced and aggregated to enable our model to effectively distinguish fine-grained intra-frame details.

## 3 METHOD

### 3.1 OVERVIEW

The architecture of our proposed method is illustrated in fig. 2, which is a novel VPS framework that imitates the behavior of a physician diagnosing highly camouflaged polyps when observing colonoscopy videos. It mainly consists of a feature extractor, a HWFA, and a decoder composed of IDPs. Specifically, the high camouflage of polyps in colonoscopy videos poses a great challenge for accurate segmentation. We propose the HWFA to solve the challenge, which decomposes features into different frequency bands and then mines subtle clues that can be used to distinguish polyps and the surroundings of polyps. Due to the dynamic nature of video, characterized by temporal changes between frames, polyp segmentation often suffers from temporal inconsistencies. Therefore, we employ the IDP in the decoder to perceive inter-frame polyp divergence, thereby mitigating the interference caused by inter-frame temporal inconsistencies and improving inter-frame polyp target tracking.

Given a video clip with T frames $\mathcal{I} \in \mathbb{R}^{T \times 3 \times H \times W}$, inspired by ZoomNeXt (Pang et al., 2024), WavePolyp first zooms in and out two auxiliary scales 0.75× and 1.25×, denoted as $\mathcal{I}^{0.75}$ and $\mathcal{I}^{1.25}$. This approach is based on the idea that different zoom levels provide distinct and complementary

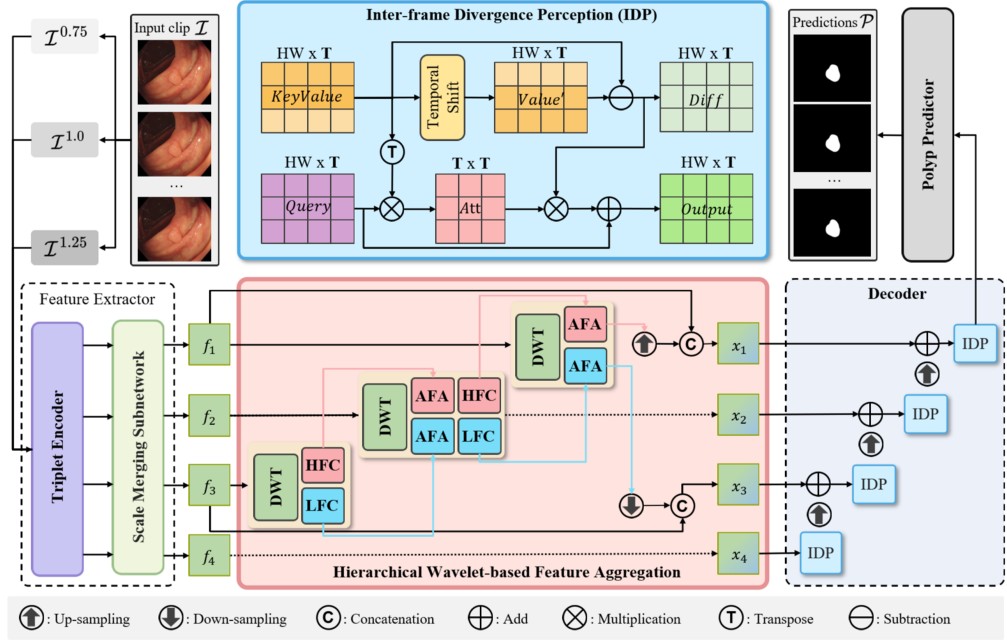

Figure 2: Overview of WavePolyp, which mainly introduces hierarchical wavelet-based feature aggregation (HWFA) and inter-frame divergence perception (IDP).

information, similar to how humans zoom in and out to gain different perspectives. These video clips of three different scales will be fed into a shared feature encoder to extract three sets of multi-level features. To aggregate the features of different scales at the 1.0x scale, these features are passed through the multi-scale feature merging Net (Pang et al., 2024), which outputs the aggregated features $\left\{ f_k \in \mathbb{R}^{T \times C \times \frac{H}{2^{k+1}} \times \frac{W}{2^{k+1}}}, \mid k \in 1, 2, 3, 4 \right\}$. The resulting features will be fed into HWFA module, decoder, and a polyp predictor. Ultimately, a clip of predictions $\mathcal{P} \in \mathbb{R}^{T \times 1 \times H \times W}$ will be generated by the polyp predictor.

## 3.2 HIERARCHICAL WAVELET-BASED FEATURE AGGREGATION

The high camouflage of polyps presents challenges for conventional feature extractors in VPS. These extractors fail to extract fine-grained discriminative features within individual frames, resulting in suboptimal performance. Existing VPS approaches focus primarily on leveraging temporal correlations between consecutive endoscopic frames to enhance intra-frame spatial representations. However, in scenarios with irregular polyp motion or low-quality video frames, such inter-frame dependency may degrade model robustness and introduce error accumulation, especially when adjacent frames are noisy or misaligned.

Discriminative features of camouflaged polyps are primarily distributed across both HF component (e.g., texture and edge), and LF component (e.g., color and illumination). This frequency-domain characteristic makes parametric-free and computationally efficient discrete wavelet transform (DWT) particularly suitable for capturing subtle variations in visually similar lesion regions. Motivated by this observation, we propose the HWFA, which enhances intra-frame feature discrimination through frequency-domain analysis. We decompose the extracted spatial features using DWT and selectively enhance the most informative components of HF and LF.

To excavate discriminative information from the decomposed features, we design a HF calculation (HFC) unit and a LF calculation (LFC) unit, as shown in fig. 3. Simultaneously, we design an ascending frequency-guided aggregation (AFA) unit to hierarchically integrate the decomposed features across different levels, rather than simply using concatenation operation based on up-sampling.

Figure 3: Details of the proposed HFC, LFC and AFA. The HFC and LFC units are designed to excavate discriminative information from the decomposed features. The AFA is designed to aggregate a lower-level feature $f_k^{HF}$ and a higher-level feature $f_{k+1}^{HF}$ under the guidance of $W_{k+1}^{HF}$. The yellow square in the AFA illustrates the application of a window-based filtering strategy that dynamically moves both vertically and horizontally.

**HFC unit.** HFC accentuates those texture-rich regions for the excavation of subtle discriminative features. Specifically, HFC receives a HF feature $f_k^{HF}$ as input, which is fed into a convolution layer, a batch normalization (BN) layer, and a residual layer for texture preservation. $f_k^{HF}$ enhanced by these layers are able to provide richer discriminative representations. We then employ channel attention and spatial attention layers to highlight noteworthy parts in spatial and channel domains. Therefore, the HFC unit is defined as:

$$W_k^{HF} = \text{CSA}(\text{BN}(\text{conv3}(f_k^{HF})) + f_k^{HF}) \tag{1}$$

where CSA denotes the layer of channel attention and spatial attention. Conv3 represents $3 \times 3$ convolution.

**LFC unit.** LF features contain more global information, which inevitably brings redundant information and interference information. Therefore, compared to HFC, the BN layer in LFC is replaced by the instance normalization (IN) (Ulyanov et al., 2017) layer, and there is an additional symmetric operation of position normalization (PN) and inverse position normalization (IPN). Specifically, LFC receives a LF feature $f_k^{LF}$ as input, which will be fed into the same layers like HFC. Note that $f_k^{LF}$ is fed into the PN layer before being fed into the convolution layer. Mean $\mu$ and variance $\sigma$ of $f_k^{LF}$ are then passed through a convolution layer to update. The IPN layer is used to restore $f_k^{LF}$ passed by the residual layer back into the original data scale according to the updated $\mu'$ and $\sigma'$. Such a design of normalization strategy (Li et al., 2019) can make the subsequent attention operation focus on cleaner global information and highlight those abnormal lesion regions from a global perspective. Therefore, the LFC unit can be formulated as follows:

$$W_k^{LF} = \text{CSA}(\text{IPN}(\text{IN}(\text{conv3}(\text{PN}(f_k^{LF})) + f_k^{LF}))) \tag{2}$$

**AFA unit.** AFA is designed to incorporate inter-feature dependencies, highlighting subtle discriminative cues. Using AFA hierarchically, aggregated features of the lowest level $f_1^{HF}$ and $f_1^{LF}$ cover multi-level enhanced feature discrimination, which enables our model to effectively distinguish fine-grained intra-frame spatial details. Specifically, taking high frequency as an example, as shown in fig. 3, AFA receives three inputs, lower-level features $f_k^{HF}$, higher-level features $f_{k+1}^{HF}$ and the guidance matrix $W_{k+1}^{HF}$ of $f_{k+1}^{HF}$, and outputs a lower-level aggregated discriminative feature $f_k^h$ that combines deep semantic information of $f_{k+1}^{HF}$ and abundant spatial details of $f_k^{HF}$. First, the down-sampled aggregated discriminative feature $f_k^{dh}$ is generated by a window-based linear model (Liu et al., 2021):

$$\left(f_k^{dh}\right)_i = \sigma_w \, \text{Down}\left(f_k^{HF}\right)_i + \mu_w, \quad \forall i \in s_w \tag{3}$$

where $\text{Down}(\cdot)$, $s_w$ and $i$ represent down-sampling, local window, and pixel point $i$. The terms $\{\sigma_w, \mu_w\}$ are the linear transformation coefficients for the pixels within the window $s_w$, which can be acquired by optimizing the subsequent objective function:

$$\min_{\sigma_w, \mu_w} \sum_{i \in s_w} \left[ (W_{k+1}^{HF})_i^2 \left((f_k^{dh})_i - ((f_{k+1}^{HF}))_i\right)^2 + \epsilon \sigma_w^2 \right] \tag{4}$$

where $\epsilon$ is responsible for constraining $\sigma_w$. Considering that pixel $i$ is covered by multiple windows, we compute the average of those window-specific coefficients to obtain the particular transformation coefficients $\{\sigma_i, \mu_i\}$ for pixel $i$. By arranging all $\{\sigma_i, \mu_i\}$ into a matrix form $\{\boldsymbol{\sigma_i}, \boldsymbol{\mu_i}\}$, Equation 3 can be reformulated as follows:

$$f_k^{dh} = \boldsymbol{\sigma_i} \odot \text{Down}\left(f_k^{HF}\right) + \boldsymbol{\mu_i} \tag{5}$$

where $\odot$ is the Hadamard product. We then up-sample $\{\sigma_i, \mu_i\}$ to obtain $\{\sigma_h, \mu_h\}$, and generate $f_k^h$ to enhance spatial details. Therefore, we can define AFA unit as:

$$f_k^h = \text{AFA}(f_k^{HF}, f_{k+1}^{HF}, W_{k+1}^{HF}) = \sigma_h \odot f_k^{HF} + \mu_h \tag{6}$$

Building on the aforementioned units, we propose HWFA, as shown in fig. 2. Specifically, we perform a DWT on $f_3$ to obtain the HF feature $f_3^{HF}$ and LF feature $f_3^{LF}$, which can be formulated as:

$$f_3^{HF} = \text{DWT}_h(f_3) \quad f_3^{LF} = \text{DWT}_l(f_3) \tag{7}$$

where $\text{DWT}_h(\cdot)$ and $\text{DWT}_l(\cdot)$ are the selection of the HF and LF sub-bands in the decomposed sub-bands obtained by applying DWT. For $f_2$ and $f_1$, a higher-level frequency feature will be aggregated into their corresponding lower-level frequency feature, which can be expressed as follows:

$$f_k^{HF} = \text{AFA}(\text{DWT}_h(f_k), f_{k+1}^{HF}, \text{HFC}(f_{k+1}^{HF})), f_k^{LF} = \text{AFA}(\text{DWT}_l(f_k), f_{k+1}^{LF}, \text{LFC}(f_{k+1}^{LF})) \tag{8}$$

Then, $f_1^{HF}$ and $f_1^{LF}$ are generated, containing multi-level aggregated discriminative information from the HF and LF components, respectively. The features $f_3$ and $f_1^{LF}$ are concatenated to enhance perception of global information, thereby preventing ignorance of excessive background information in higher-level features. The features $f_1$ and $f_1^{HF}$ are concatenated to enhance the ability of $f_1$ to capture fine details, such as small objects, detailed edges, and intricate textures. Finally, the enhanced features $x_1$, $x_2$, $x_3$, and $x_4$ will be obtained, which can be written as:

$$x_4 = f_4, x_3 = \text{conv}_{1\times1}(\text{Concat}(\text{Down}(f_1^{LF}), f_3)) \tag{9}$$

$$x_2 = f_2, x_1 = \text{conv}_{1\times1}(\text{Concat}(\text{Up}(f_1^{HF}), f_1)) \tag{10}$$

where $\text{Concat}(\cdot)$, $\text{Up}(\cdot)$ and $\text{conv}_{1\text{x}1}(\cdot)$ denote concatenation, up-sampling, and $1 \times 1$ convolution.

### 3.3 INTER-FRAME DIVERGENCE PERCEPTION

Polyps in colonoscopy videos may have significant inter-frame divergence in shape, location, size, and boundary, which manifests as temporal inconsistencies between frames. In addition, polyp tissues are prone to non-rigid deformations caused by intestinal peristalsis and camera jitter. Relying solely on intra-frame feature often fails to distinguish artifacts caused by non-rigid deformations from realistic polyp structural changes, and neglects valuable temporal cues present across frames.

To enhance the model's ability to capture temporal dynamics and distinguish meaningful inter-frame variations, we introduce IDP blocks. These blocks improve the tracking of polyp targets among consecutive frames by leveraging our proposed inter-frame divergence perception mechanism. The core idea is to explicitly model the temporal divergence between adjacent frames. Specifically, as shown in the IDP block of fig. 2, given the visual features $Q$ and $KV$, which are obtained from $x_k$ through convolution followed by flattening, we first perform the temporal shift (Lin et al., 2019). After shifting, the feature map of the starting time step in the video clip is moved to the end, while the remaining feature maps move forward by one position. Consequently, we derive the inter-frame divergence $Diff = \text{Shift}(V) - V$, and modulate it via a learnable projection matrix $W_V$, allowing the network to focus on regions with significant temporal changes. To enhance the motion cues of the object of interest across frames, we apply an inter-frame divergent attention operation, given by $(W_V(\text{Shift}(V) - V))\text{Softmax}(\frac{K^T Q}{\sqrt{\text{HW}}})$. Notably, the attention mechanism operates along the time dimension. The resulting weight matrix, denoted as Att, is a T×T matrix, representing the interactions and comparisons between each time step and all others. The product of Att and Diff serves to identify the most significant information within the time series, thereby highlighting the inter-frame divergence essential for tracking polyp locations. Since the computation is conducted along the time dimension rather than the channel dimension, the computational complexity remains relatively low. Furthermore, two T × 3 × 3 convolutional layers are applied to the temporal cues to fully diffuse the inter-frame divergence information, which is then added to the original visual features $Q$, which we call it inter-frame divergence diffusion. The final output is computed as:

$$O = \text{Conv}(W_V(\text{Shift}(V) - V)\text{Softmax}(\frac{K^T Q}{\sqrt{\text{HW}}}) + Q) \tag{11}$$

where $Q, K \in R^{HW \times T}$. $\text{Conv}(\cdot)$, $W_V$, $\text{Shift}(\cdot)$ are two T × 3 × 3 convolutional layers, a T × T learnable matrix and the temporal shift operation like (Lin et al., 2019).

Table 1: Quantitative comparison with different SOTA methods on SUN-SEG and CVC-612.

| Model | Backbone | SUN-SEG-Easy | | | | SUN-SEG-Hard | | | | CVC-612 | | | |
|---|---|---|---|---|---|---|---|---|---|---|---|---|---|
| | | $S_\alpha$ | $E_\phi^{mn}$ | $F_\beta^w$ | Dice | $S_\alpha$ | $E_\phi^{mn}$ | $F_\beta^w$ | Dice | $S_\alpha$ | $E_\phi^{mn}$ | $F_\beta^w$ | Dice |
| SLT-Net | PVTv2-B5 | 90.39 | 93.75 | 84.35 | 87.15 | 88.06 | 92.05 | 80.31 | 83.55 | 94.61 | 97.70 | 92.06 | 92.96 |
| ZoomNeXt | PVTv2-B5 | 89.81 | 92.25 | 84.64 | 87.55 | 88.51 | 91.34 | 82.21 | 85.22 | 94.54 | 97.53 | 90.91 | 92.73 |
| AutoSAM | ViT-B | 86.28 | 91.69 | 78.25 | 81.27 | 83.59 | 89.91 | 73.08 | 77.25 | 90.56 | 96.19 | 87.68 | 88.34 |
| WeakPolyp | PVTv2-B5 | 90.51 | 93.72 | 84.89 | 87.57 | 90.19 | 93.77 | 83.74 | 86.73 | 91.51 | 95.18 | 88.74 | 89.07 |
| PNS+ | Res2Net-50 | 85.75 | 86.11 | 76.14 | 81.91 | 83.98 | 85.68 | 72.75 | 79.32 | 94.49 | 96.44 | 89.05 | 92.54 |
| MAST | PVTv2-B2 | 87.91 | 92.87 | 81.40 | 84.44 | 87.44 | 92.82 | 80.27 | 83.79 | 93.54 | 96.07 | 89.93 | 90.12 |
| SALI | PVTv2-B5 | 89.45 | 93.01 | 83.73 | 86.07 | 89.19 | 93.21 | 83.05 | 85.54 | 94.41 | 97.12 | 92.16 | 93.01 |
| VP-SAM | ViT-B | 90.49 | 93.68 | 85.64 | 88.19 | 90.03 | 93.74 | 83.30 | 86.94 | 94.68 | 97.83 | 92.51 | 93.45 |
| **Ours** | PVTv2-B5 | **90.93** | **94.15** | **86.74** | **88.96** | **90.28** | **93.96** | **85.17** | **87.55** | **95.24** | **98.61** | **93.57** | **94.36** |

Figure 4: (a) Visualization of module ablation on test set SUN-SEG-Hard. (b-c) t-SNE visualization of features. Red represents lesion regions, while blue represents the opposite.

The decoder stacks four IDP in a coarse-to-fine cascade: starting from the deepest feature $x_4$, each IDP temporally enhances the current feature via IDP, upsamples it, and residually fuses it with the feature of the next finer scale; after four such iterations the final representation is obtained and fed to the polyp predictor to generate prediction result.

## 3.4 Loss Functions

We apply binary cross-entropy loss $L_{bce}$ like (Ji et al., 2022) to guide the convergence process of the model. Moreover, the uncertainty awareness loss $L_{ual}$ like (Pang et al., 2024) becomes another part of our loss in order to force the model to increase "confidence" in decision making and aggravate the penalty for fuzzy prediction. We can formulate the total loss with two components as follows:

$$L_{total} = L_{bce}(P_t, G_t) + \lambda L_{ual}(P_t, G_t) \tag{12}$$

where $\lambda$ is the balance coefficient and follows the adjustment strategy in Pang et al. (2024), while $P_t$ and $G_t$ denote the prediction and the corresponding ground truth (GT), respectively.

## 4 Experiment

### 4.1 Experiment Setup

**Datasets.** We evaluate our method on two public video-based datasets: SUN-SEG (Ji et al., 2022), and CVC-612 (Bernal et al., 2015). (1) SUN-SEG includes 49, 136 frames from 285 clips, which consists of three subsets: training set (112 clips / 19, 544 frames), test set SUN-SEG-Easy (119 clips / 17,070 frames) and test set SUN-SEG-Hard (54 clips / 12,544 frames). (2) CVC-612 contains 612 frames from 31 colonoscopy clips with a resolution of 384 × 288.

**Evaluation Metrics.** For comprehensive and fair comparison, we employ four metrics to evaluate the results, including Dice, structure-measure ($S_\alpha$) (Fan et al., 2017), enhanced-alignment measure ($E_\phi^{mn}$) (Fan et al., 2021), and weighted F-measure ($F_\beta^w$) (Margolin et al., 2014) , similar to previous works (Ji et al., 2022).

**Implementation Details.** Our proposed method is implemented using PyTorch. All frames are uniformly resized to 352×352, while clip length T and channel dimension C are respectively set to 5 and 64. The entire model is trained for 30 epochs with a batch size of 2 in an end-to-end manner on a single RTX 3090 GPU, using Adam with betas = (0.9, 0.999). The learning rates of backbone

Table 2: Ablation studies conducted on the SUN-SEG test set to evaluate the two core components of WavePolyp and their internal designs. Norm: Normalization strategy of LFC. IDA: Inter-frame divergence attention. IDD: Inter-frame divergence diffusion.

| HWFA | | | | IDP | | | SUN-SEG-Easy | | SUN-SEG-Hard | |
|---|---|---|---|---|---|---|---|---|---|---|
| HFC | LFC | Norm | AFA | Shift | IDA | IDD | $S_\alpha$ | Dice | $S_\alpha$ | Dice |
| | | | | | | | 89.12 | 87.51 | 88.84 | 86.12 |
| | | | | ✓ | ✓ | ✓ | 90.07 | 87.69 | 90.09 | 86.65 |
| | ✓ | ✓ | ✓ | ✓ | ✓ | ✓ | 90.31 | 88.06 | 89.75 | 87.12 |
| ✓ | | | ✓ | ✓ | ✓ | ✓ | 89.81 | 87.29 | 89.79 | 86.85 |
| ✓ | ✓ | | ✓ | ✓ | ✓ | ✓ | 90.21 | 87.76 | 89.96 | 87.01 |
| ✓ | ✓ | ✓ | | ✓ | ✓ | ✓ | 87.92 | 87.39 | 87.75 | 85.55 |
| ✓ | ✓ | ✓ | ✓ | | | | 90.03 | 87.62 | 89.45 | 86.76 |
| ✓ | ✓ | ✓ | ✓ | | ✓ | ✓ | 90.26 | 88.01 | 89.74 | 87.11 |
| ✓ | ✓ | ✓ | ✓ | ✓ | ✓ | | 90.57 | 88.36 | 90.06 | 87.12 |
| ✓ | ✓ | ✓ | ✓ | ✓ | ✓ | ✓ | **90.93** | **88.96** | **90.28** | **87.55** |

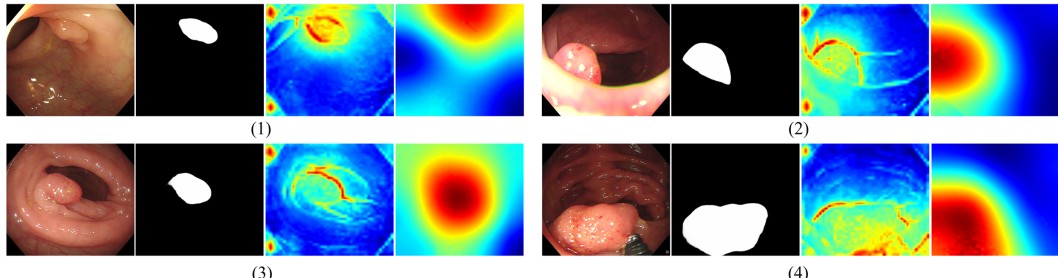

(1)  (2)

(3)  (4)

Figure 5: Visual analysis of frequency-based feature disentanglement. From left to right: (a) Original Frame, (b) Prediction Mask, (c) High-Frequency Features ($f_1^{HF}$), and (d) Low-Frequency Features ($f_1^{LF}$).

and the rest are initialized to 1e-5 and 2e-5, and both decrease by 2% every epoch. Unless otherwise specified, PVTv2-b5 (Wang et al., 2022b) is used as our backbone for all experiments, which is pre-trained on ImageNet. For SUN-SEG, we separate 20% from the training set as the validation set. For CVC-612, we split the training set, validation set, and test set with a ratio of 6 : 2 : 2.

## 4.2 ABLATION STUDIES

**Effectiveness of Components.** All ablation experiments are conducted on SUN-SEG. As shown in table 2, the performance degradation observed when removing either HWFA or IDP suggests that both components are essential for improving segmentation performance in VPS. Notably, when HWFA and IDP are applied individually, the improvement in the Dice metric is more pronounced on the hard test set, indicating their strong adaptability to challenging samples. Moreover, the performance gain from using each component separately is less significant than when they are combined, confirming the complementary nature of intra-frame discriminative feature excavation and inter-frame divergence perception. In addition, the individual units HFC, LFC and AFA also demonstrate improved robustness against difficult cases. The standardization strategy of LFC has demonstrated its effectiveness. Significant performance drops are observed when AFA is excluded, even falling below the baseline. This is because HFC and LFC strongly enhance the high-frequency textures and low-frequency structures. Without the adaptive spatial alignment provided by AFA, naively fusing these sharpened features across different scales amplifies the inherent spatial misalignments, introducing noise and artifacts that are more detrimental than the unenhanced features in the baseline. To explicitly validate the effectiveness of the HWFA module, we visualized the final aggregated frequency features, $f_1^{HF}$ and $f_1^{LF}$ in fig. 5. These features represent the culmination of hierarchical

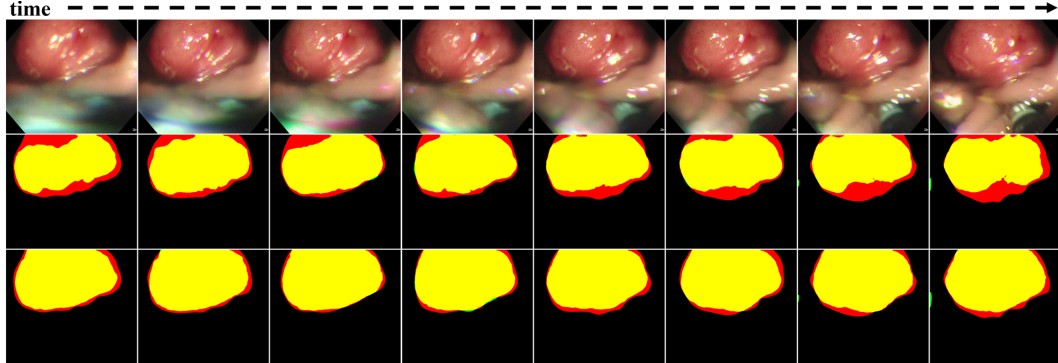

Figure 6: Impact of IDP on temporal stability. Row 2: w/o IDP; Row 3: Ours. The IDP module ensures consistent tracking and suppresses flickering. (Red: GT, Green: Pred, Yellow: Overlap).

aggregation and are used to refine the main features $f_1$ and $f_3$, respectively. As observed in the visualization, $f_1^{HF}$ (HF output) sharply highlights the polyp's boundaries and surface textures. This aggregated high-frequency cue is fused into the shallow layer ($x_1$) to enhance the detail preservation of the camouflaged object. $f_1^{LF}$ (LF output) captures the holistic lesion body and structural context. This aggregated low-frequency cue is fused into the deep layer ($x_3$) to prevent the loss of global semantic information. This complementary behavior confirms that HWFA successfully disentangles and aggregates discriminative cues for targeted feature refinement. Lastly, the three distinct components of the IDP have demonstrated their indispensability. Meanwhile, we perform frame-by-frame visualization in fig. 6, which shows that excluding the IDP module leads to mask flickering and unstable boundaries (second row), while our IDP-equipped model ensures smooth and temporally consistent segmentation (third row). For more extensive visualization results, please refer to section A.5.

Furthermore, we provide a visual comparison to illustrate the contributions of key components in WavePolyp. As shown in fig. 4, IDP effectively alleviates segmentation discontinuities caused by drastic variations in polyp size, shape, and position. Meanwhile, HWFA addresses the challenge of high camouflage in polyps, leading to clearer boundaries and higher confidence. When HWFA and IDP are combined, the network achieves superior segmentation performance, characterized by sharper boundaries, more precise localization, accurate size and shape estimation, and higher confidence in predictions.

**Effect of Clip Length.** As shown in fig. 8, we perform comprehensive evaluations of varying clip lengths, with experimental results demonstrating optimal performance at a clip length of 5. We analyze that shorter clips inadequately capture inter-frame divergence, leading to localization failures during rapid polyp morphological changes. In contrast, longer clips introduce excessive divergence between temporally distant frames that degrade model decision making.

## 4.3 COMPARISON WITH STATE-OF-THE-ART METHODS

We compare WavePolyp with some SOTA methods, including SLT-Net (Cheng et al., 2022), Zoom-NeXt (Pang et al., 2024), AutoSAM (Shaharabany et al., 2023), WeakPolyp (Wei et al., 2023), PNS+ (Ji et al., 2022), MAST (Chen et al., 2024), SALI (Hu et al., 2024) and VP-SAM (Fang et al., 2024). These methods include two natural video segmentation models (SLT-Net and ZoomNeXt), two IPS methods (AutoSAM and WeakPolyp), and four VPS methods. To ensure fairness, we obtain the segmentation results of these methods using their publicly available implementations.

The quantitative comparison between our method and above SOTA methods on SUN-SEG and CVC-612 is presented in table 1. The results demonstrate that our method outperforms other SOTA methods across all metrics, illustrating its robustness. VP-SAM achieves performance close to ours but relies on a single point prompt for each frame, which is impractical to obtain in clinical applications. In addition, we provide a qualitative comparison with SOTA methods. As shown in fig. 7, our method yields more accurate segmentation results when dealing with highly camouflaged polyps and significant inter-frame divergence. Compared to other methods, it achieves superior polyp lo-

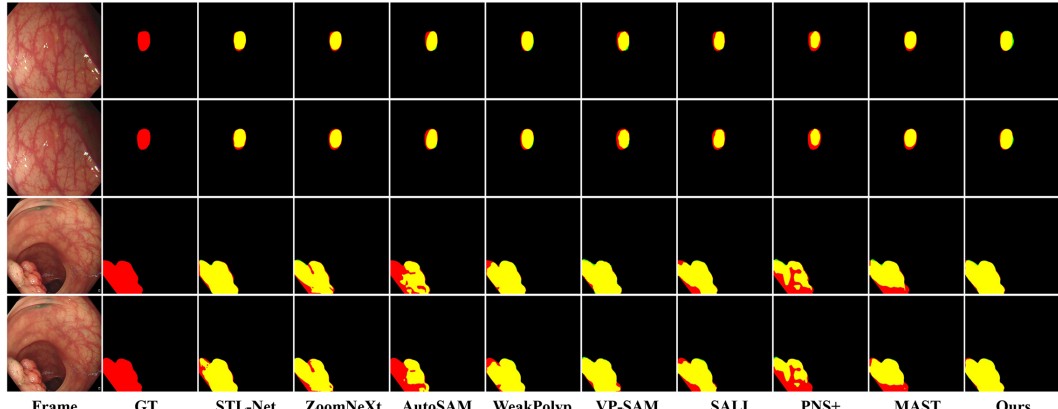

Figure 7: Visual comparison of our proposed method with different SOTA methods on the SUN-SEG-Easy and SUN-SEG-Hard test sets. Red, green and yellow represent the GT, prediction and their overlapping regions, respectively.

Table 3: Performance vs. efficiency on SUN-SEG-Hard based on an RTX 3090 GPU and batch size equal to 1.

| Methods | $S_\alpha$ | Dice | Params | GFLOPs | FPS |
|---|---|---|---|---|---|
| SLT-Net | 88.06 | 83.55 | 82.39M | 87.81 | 5.08 |
| ZoomNeXt | 88.51 | 85.22 | 84.78M | 102.32 | 22.48 |
| PNS+ | 83.98 | 79.32 | 9.79M | 46.01 | 75.21 |
| MAST | 87.44 | 83.79 | 25.69M | 21.02 | 21.69 |
| SALI | 89.19 | 85.54 | 82.73M | 58.15 | 7.93 |
| VP-SAM | 90.03 | 86.94 | 140.27M | 156.86 | 12.74 |
| **WavePolyp** | **90.28** | **87.55** | **86.63M** | **114.88** | **23.04** |

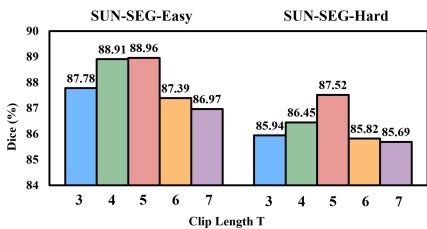

Figure 8: Ablation studies of clip length on SUN-SEG.

calization and clearer boundary delineation, demonstrating the effectiveness of HWFA in excavating intra-frame discriminative features and IDP in perceiving inter-frame polyp divergence.

## 4.4 DISCUSSIONS AND LIMITATIONS

Although our experiments are conducted solely on colonoscopy video datasets, we believe that our method is general enough to be applied to other datasets with similar challenges. Furthermore, as shown in table 3, we conduct performance-efficiency comparison with video-based SOTA methods. Our method achieves significant performance improvements near real-time inference speed (23.04 FPS). However, our method still has some limitations. As shown in fig. 9, missing details (a-b), distorted lighting and focus (c-d) and overlapping intestinal walls (e-f) may limit our method.

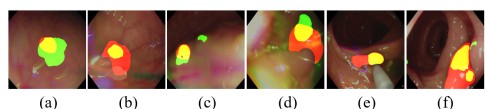

Figure 9: Failure cases. Red, green and yellow represent the GT, prediction, and their overlapping regions, respectively.

## 5 CONCLUSION

In this paper, we propose a novel network WavePolyp, consisting of two innovative components: a HWFA module and IDP blocks, which improves the segmentation performance from both intra-frame and inter-frame perspectives. The HWFA allows WavePolyp to identify highly camouflaged polyps by excavating intra-frame discriminative features. Additionally, the IDP assists in accurate polyp tracking to address temporal inconsistency. Extensive experimental results on SUN-SEG and CVC-612 demonstrate the effectiveness of our proposed method.

ACKNOWLEDGMENTS

This work was supported partly by National Natural Science Foundation of China (No. 62273241), Natural Science Foundation of Guangdong Province, China (No. 2024A1515011946), the Shenzhen Research Foundation for Basic Research, China (No. JCYJ20250604181940054), and a grant under Collaborative Research with World-leading Research Groups scheme of The Hong Kong Polytechnic University (project no. G-SACF).

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

# A APPENDIX

## A.1 DETAILS OF CHANNEL AND SPATIAL ATTENTION (CSA)

In the proposed HFC and LFC units, we employ a sequential channel attention (CA) and spatial attention (SA) module, denoted as CSA, to selectively emphasize informative features in both channel and spatial dimensions. The process can be formulated as:

$$F' = \mathcal{M}_c(F) \otimes F, \quad F'' = \mathcal{M}_s(F') \otimes F', \tag{13}$$

where $F \in \mathbb{R}^{C \times H \times W}$ is the input feature, $\mathcal{M}_c$ and $\mathcal{M}_s$ denote the channel and spatial attention maps, respectively, and $\otimes$ represents element-wise multiplication.

**Channel Attention (CA).** Different from the standard MLP-based approach, we adopt a more efficient 1D convolution strategy to capture local cross-channel interactions. Furthermore, considering the distinct characteristics of wavelet-decomposed features, we employ adaptive pooling strategies. For LFC, which represent smooth global structures, we use *Global Average Pooling* (GAP) to aggregate background information. For HFC, which contain sharp edges and textures, we use *Global Max Pooling* (GMP) to preserve the most discriminative texture signals.

The channel attention map $\mathcal{M}_c(F) \in \mathbb{R}^{C \times 1 \times 1}$ is computed as:

$$\mathcal{M}_c(F) = \sigma \left( \text{Conv1D}_k \left( \text{Pool}(F) \right) \right), \tag{14}$$

where $\sigma$ is the Sigmoid activation function, and $\text{Conv1D}_k$ represents a 1D convolution with a kernel size of $k = 3$. $\text{Pool}(\cdot)$ denotes GAP for LFC units and GMP for HFC units.

**Spatial Attention (SA).** The spatial attention module focuses on highlighting where the informative regions are. We employ a bottleneck convolutional structure to compute the spatial attention map $\mathcal{M}_s(F') \in \mathbb{R}^{1 \times H \times W}$. The refined feature $F'$ is first reduced in channel dimension to extract local spatial contexts and then mapped to a single channel:

$$\mathcal{M}_s(F') = \sigma \left( \text{Conv}_2 \left( \delta \left( \text{Conv}_1(F') \right) \right) \right), \tag{15}$$

where $\text{Conv}_1$ is a $3 \times 3$ convolution reducing channels from $C$ to $C/4$, $\delta$ is the ReLU activation function, and $\text{Conv}_2$ is a $3 \times 3$ convolution projecting channels from $C/4$ to $1$. This design effectively suppresses background noise while highlighting the polyp regions.

## A.2 MORE DETAILS ON AFA

In low-frequency components, AFA receives three inputs, lower-level features $f_k^{LF}$, higher-level features $f_{k+1}^{LF}$ and the guidance matrix $W_{k+1}^{LF}$ of $f_{k+1}^{LF}$, and outputs a lower-level aggregated feature $f_k^l$, which hierarchically integrates the enhanced discriminative features of highly camouflaged polyps. The aggregated feature $f_k^l$ can be generated by minimizing the subsequent objective function:

$$\min_{\sigma_w, \mu_w} \sum_{i \in s_w} \left[ (W_{k+1}^{LF})_i^2 \left( (f_k^{dl})_i - ((f_{k+1}^{LF}))_i \right)^2 + \epsilon \sigma_w^2 \right] \tag{16}$$

where

$$\left( f_k^{dl} \right)_i = \sigma_w \, \text{Down} \left( f_k^{LF} \right)_i + \mu_w, \quad \forall i \in s_w \tag{17}$$

where $\text{Down}(\cdot)$, $s_w$ and $i$ represent down-sampling, local window centered by pixel $w$ and pixel point $i$. $\epsilon$ is responsible for constraining $\sigma_w$. Similar to high-frequency components, by averaging,

matrixing and up-sampling the window-based coefficients, we get the final transformation coefficients $\{\boldsymbol{\sigma_h^l}, \boldsymbol{\mu_h^l}\}$. Therefore, the aggregated feature $f_k^l$ can be obtained as follows:

$$
\begin{aligned}
f_k^l &= \text{AFA}(f_k^{LF}, f_{k+1}^{LF}, W_{k+1}^{LF}) \\
&= \boldsymbol{\sigma_h^l} \odot f_k^{LF} + \boldsymbol{\mu_h^l}
\end{aligned}
\tag{18}
$$

where $\odot$ is the Hadamard product.

### A.3 MORE ABLATION STUDIES

**Design of HWFA.** As shown in fig. 2, our HWFA design begins with hierarchical aggregation starting from $f_3$. This design choice is supported by experimental results. In table 4, we present the results of ablation experiments with four different design variations. Experiment 1 applies DWT, HFC, and LFC only to $f_1$, omitting AFA. Experiment 2 employs level-wise fusion starting from $f_2$. Experiment 3 uses level-wise fusion starting from $f_3$. Experiment 4 starts level-wise fusion from $f_4$, with the note that this experiment was conducted at a resolution of 384×384, as DWT requires an even resolution. In the 352×352 setting, the resolution of $f_4$ is 11×11. Therefore, we chose the design from Experiment 3 as the final HWFA design.

Table 4: Ablation studies conducted on the SUN-SEG test set to evaluate the design of HWFA.

| Design | SUN-SEG-Easy | | SUN-SEG-Hard | |
|---|---|---|---|---|
| | $S_\alpha$ | Dice | $S_\alpha$ | Dice |
| 1 | 88.91 | 86.53 | 88.67 | 85.94 |
| 2 | 90.14 | 88.52 | 89.75 | 87.03 |
| 3 | **90.93** | **88.96** | **90.28** | **87.55** |
| 4 | 90.65 | 88.84 | 90.13 | 87.47 |

**Channel Dimension.** We compared different channel dimensions, and as shown in table 6, the best performance is achieved when C = 64.

**Learning Rate.** We also tried different base learning rates (baselr) of backbone. The base learning rates of the rest components is twice that of the former. The results are shown in table 5. When baselr = 1e-5, the segmentation performance is the best.

Table 5: Performance of different base learning rates on the SUN-SEG test set.

| baselr | SUN-SEG-Easy | | SUN-SEG-Hard | |
|---|---|---|---|---|
| | $S_\alpha$ | Dice | $S_\alpha$ | Dice |
| 5e-6 | 90.75 | 88.12 | 89.54 | 87.17 |
| 1e-5 | **90.93** | **88.96** | **90.28** | **87.55** |
| 5e-5 | 90.65 | 88.45 | 89.75 | 87.16 |
| 1e-4 | 90.27 | 87.76 | 89.37 | 86.97 |

Table 6: Performance of different channel dimensions on the SUN-SEG test set.

| C | SUN-SEG-Easy | | SUN-SEG-Hard | |
|---|---|---|---|---|
| | $S_\alpha$ | Dice | $S_\alpha$ | Dice |
| 32 | 90.64 | 88.05 | 89.17 | 87.01 |
| 64 | **90.93** | **88.96** | **90.28** | **87.55** |
| 128 | 90.77 | 88.69 | 89.95 | 87.41 |
| 256 | 90.35 | 88.71 | 90.03 | 87.44 |

### A.4 MORE QUANTITATIVE COMPARISON RESULTS

We employ six metrics to evaluate the results, including Dice, structure-measure ($S_\alpha$) Fan et al. (2017), enhanced-alignment measure ($E_\phi^{mn}$) Fan et al. (2021), sensitivity ($Sen$), F-measure ($F_\beta^{mn}$) Achanta et al. (2009), and weighted F-measure ($F_\beta^w$) Margolin et al. (2014) , similar to previous works Ji et al. (2022); Hu et al. (2024). The test set SUN-SEG-Easy has two subsets: SUN-SEG-Easy-Seen and SUN-SEG-Easy-Unseen, where Seen represents the visible case that divides one case into two parts for training and testing. Similarly, the test set SUN-SEG-Hard has two responding subsets. The more detailed quantitative comparison results are shown in table 7, which demonstrates that our method outperforms other state-of-the-art methods in most metrics, illustrating the robustness and superiority of our method.

In addition, to verify the cross-dataset generalizability of our proposed method, we conducted an additional experiment where the model was trained on the SUN-SEG dataset and directly evaluated on the CVC-612 dataset (including train and validation sets) without any fine-tuning. As shown in

Table 7: More detailed quantitative comparison with different SOTA methods on SUN-SEG test sets.

| | Method | SUN-SEG-Easy | | | | | | SUN-SEG-Hard | | | | | |
|---|---|---|---|---|---|---|---|---|---|---|---|---|---|
| | | $S_\alpha$ | $E_\phi^{mn}$ | $F_\beta^w$ | $F_\beta^{mn}$ | Sen | Dice | $S_\alpha$ | $E_\phi^{mn}$ | $F_\beta^w$ | $F_\beta^{mn}$ | Sen | Dice |
| Seen | SLT-Net | 93.46 | 96.23 | 89.11 | 91.35 | 89.94 | 91.20 | 90.28 | 93.90 | 84.01 | 87.02 | 85.81 | 87.04 |
| | ZoomNeXt | 94.00 | 96.34 | 90.86 | 92.84 | 89.73 | 92.40 | 90.78 | 93.72 | 86.07 | 88.77 | 84.72 | 88.09 |
| | AutoSAM | 90.19 | 94.22 | 83.20 | 85.77 | 89.08 | 86.49 | 85.99 | 91.42 | 76.84 | 80.26 | 85.69 | 81.30 |
| | WeakPolyp | 93.64 | 96.61 | 89.42 | 91.24 | 91.52 | 91.61 | 92.22 | 95.68 | 86.78 | 89.51 | 88.35 | 89.49 |
| | PNS+ | 91.35 | 92.40 | 84.72 | 87.86 | 83.83 | 88.34 | 88.46 | 92.06 | 80.29 | 84.93 | 78.07 | 85.14 |
| | MAST | 92.58 | 96.29 | 87.84 | 90.91 | 88.18 | 89.98 | 89.24 | 94.28 | 83.29 | 87.76 | 83.24 | 86.49 |
| | SALI | 93.86 | 96.52 | 90.51 | 92.14 | 91.06 | 92.16 | 90.59 | 94.08 | 85.65 | 88.17 | 86.58 | 87.68 |
| | VP-SAM | 94.35 | 96.94 | 90.82 | 92.77 | 90.74 | 92.77 | 92.00 | 95.54 | 86.51 | 88.59 | 87.74 | 89.87 |
| | **Ours** | **94.47** | **97.26** | **91.89** | **93.72** | 90.91 | **93.05** | **92.32** | 95.78 | **88.46** | **91.14** | 87.69 | **90.31** |
| Unseen | SLT-Net | 87.32 | **91.27** | 79.59 | 83.59 | **79.07** | 83.10 | 85.84 | 90.20 | 76.61 | 80.64 | 77.59 | 80.06 |
| | ZoomNeXt | 85.62 | 88.16 | 78.42 | 82.99 | 74.38 | 82.70 | 86.24 | 88.96 | 78.35 | 82.41 | 77.14 | 82.35 |
| | AutoSAM | 82.37 | 89.16 | 73.30 | 77.78 | 74.74 | 76.05 | 81.19 | 88.40 | 69.32 | 73.41 | 75.75 | 73.20 |
| | WeakPolyp | 87.38 | 90.81 | 80.36 | 84.41 | 78.33 | 83.53 | 88.16 | 91.86 | 80.70 | 84.34 | 81.35 | 83.97 |
| | PNS+ | 80.15 | 79.82 | 67.56 | 73.05 | 63.04 | 75.48 | 79.50 | 79.30 | 65.21 | 70.99 | 62.33 | 73.50 |
| | MAST | 83.24 | 89.45 | 74.96 | 80.96 | 73.97 | 78.90 | 85.64 | 91.36 | 77.25 | 82.12 | 79.04 | 81.09 |
| | SALI | 85.04 | 89.50 | 76.95 | 81.11 | 75.56 | 79.98 | 87.79 | **92.34** | 80.45 | 83.87 | **82.18** | 83.40 |
| | VP-SAM | 86.63 | 90.42 | 80.46 | 83.64 | 77.65 | 83.61 | 88.06 | 91.94 | 80.09 | 82.94 | 81.71 | 84.01 |
| | **Ours** | **87.39** | 91.04 | **81.59** | **85.87** | 78.45 | **84.87** | **88.24** | 92.14 | **81.88** | **85.63** | 81.41 | **84.79** |

Table 8: Cross-dataset generalization performance. The model was trained on SUN-SEG and tested directly on CVC-612.

| Model | Backbone | CVC-612 | | | | | |
|---|---|---|---|---|---|---|---|
| | | $S_\alpha$ | $E_\phi^{mn}$ | $F_\beta^w$ | $F_\beta^{mn}$ | Sen | $Dice$ |
| SLT-Net | PVTv2-B5 | 89.58 | 91.62 | 84.85 | 88.35 | 81.12 | 86.29 |
| ZoomNeXt | PVTv2-B5 | 89.51 | 91.57 | 84.72 | 88.21 | 80.97 | 86.24 |
| AutoSAM | ViT-B | 85.35 | 89.85 | 78.92 | 83.56 | 75.12 | 80.45 |
| WeakPolyp | PVTv2-B5 | 88.74 | 91.09 | 82.77 | 86.24 | 81.45 | 84.79 |
| PNS+ | Res2Net-50 | 87.52 | 90.45 | 81.24 | 85.10 | 78.50 | 83.65 |
| MAST | PVTv2-B2 | 85.89 | 90.21 | 79.39 | 84.28 | 75.93 | 81.46 |
| SALI | PVTv2-B5 | 89.62 | 91.14 | 84.91 | 87.54 | 81.07 | 86.32 |
| VP-SAM | ViT-B | 89.68 | 92.25 | 85.42 | 88.85 | 81.55 | 86.95 |
| **Ours** | PVTv2-B5 | **89.74** | **93.07** | **85.89** | **89.31** | **81.87** | **87.36** |

table 8, our method achieves competitive performance even on unseen data distributions, demonstrating its strong generalization capability.

## A.5 MORE VISUAL COMPARISON RESULTS

We demonstrate a qualitative comparison between our method and other state-of-the-art methods on the CVC-612 dataset, as shown in fig. 10. To comprehensively demonstrate the effectiveness of the Inter-frame Divergence Perception (IDP) module, particularly its ability to maintain temporal consistency under rapid motion, we provide two additional visual ablation examples in this section. As shown in fig. 11 and fig. 12, the IDP module significantly reduces segmentation jitter and ensures robust tracking compared to the baseline. In addition, to demonstrate the superior temporal consistency of our method, we provide visual comparisons of two video sequences, each consisting of seven consecutive frames, as shown in fig. 13 and fig. 14. The results demonstrate that our method excels in both distinguishing polyps from the background and ensuring excellent temporal consistency.

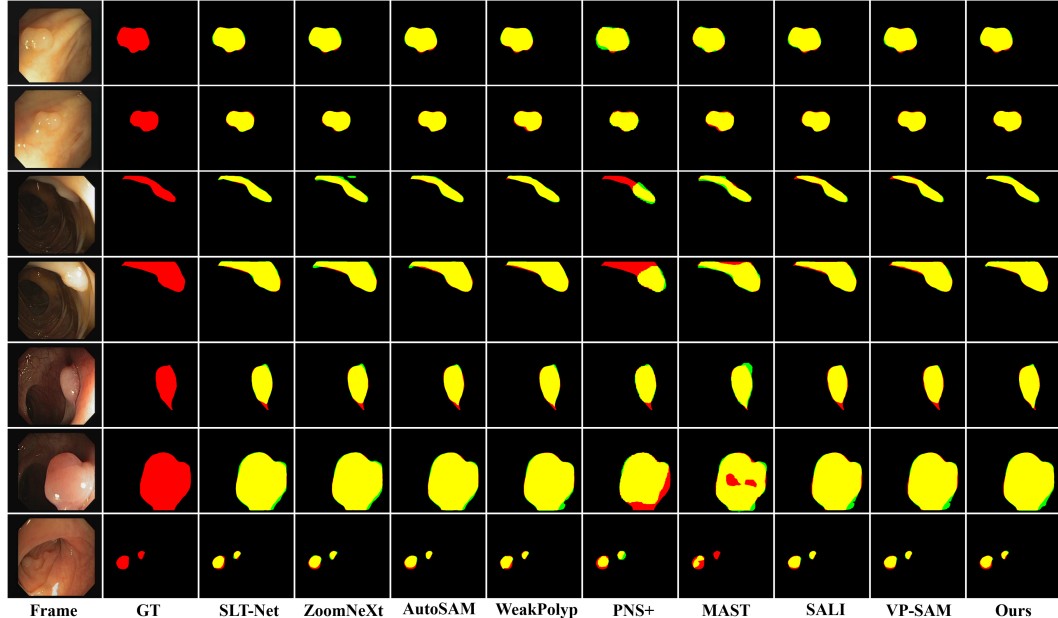

Figure 10: More visual comparison results of our method with other state-of-the-art methods on the CVC-612 test set. Red, green and yellow represent the GT, prediction and their overlapping regions, respectively. Please zoom in for more details.

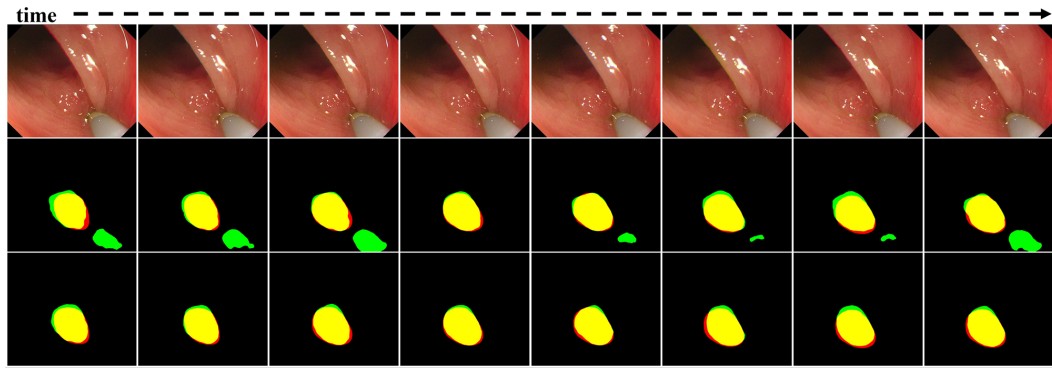

Figure 11: More visual ablation of IDP on SUN-SEG. Row 2: w/o IDP; Row 3: Ours. (Red: GT, Green: Pred, Yellow: Overlap).

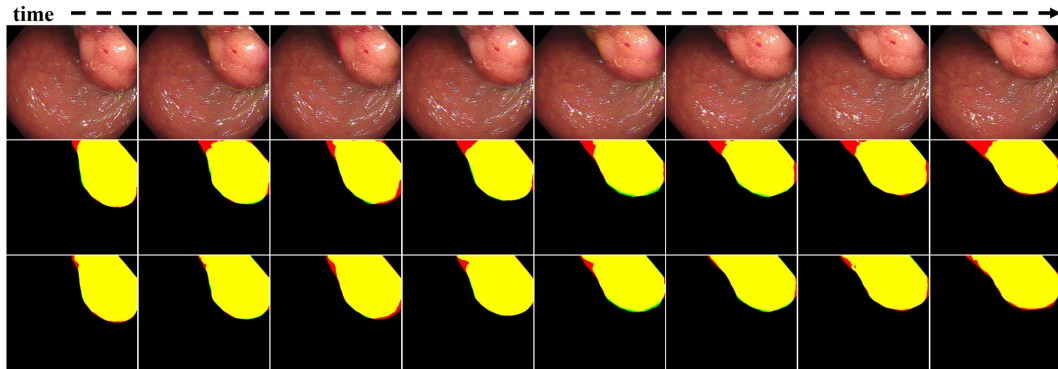

Figure 12: More visual ablation of IDP on SUN-SEG. Row 2: w/o IDP; Row 3: Ours. (Red: GT, Green: Pred, Yellow: Overlap).

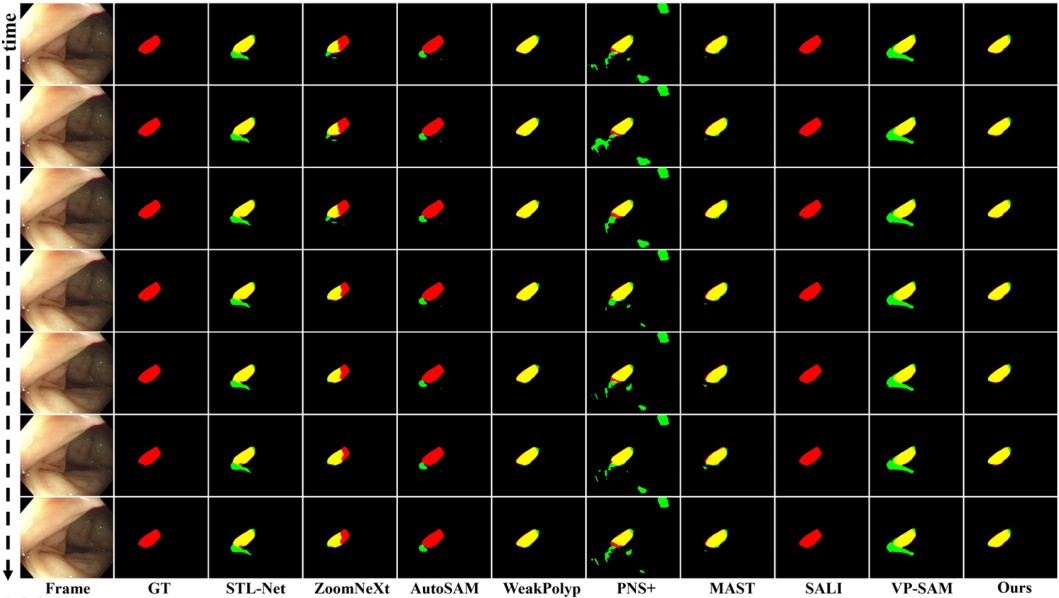

Figure 13: Frame-by-frame visual comparison of 7-frame video sequence selected from the SUN-SEG dataset. (Red: GT, Green: Pred, Yellow: Overlap).

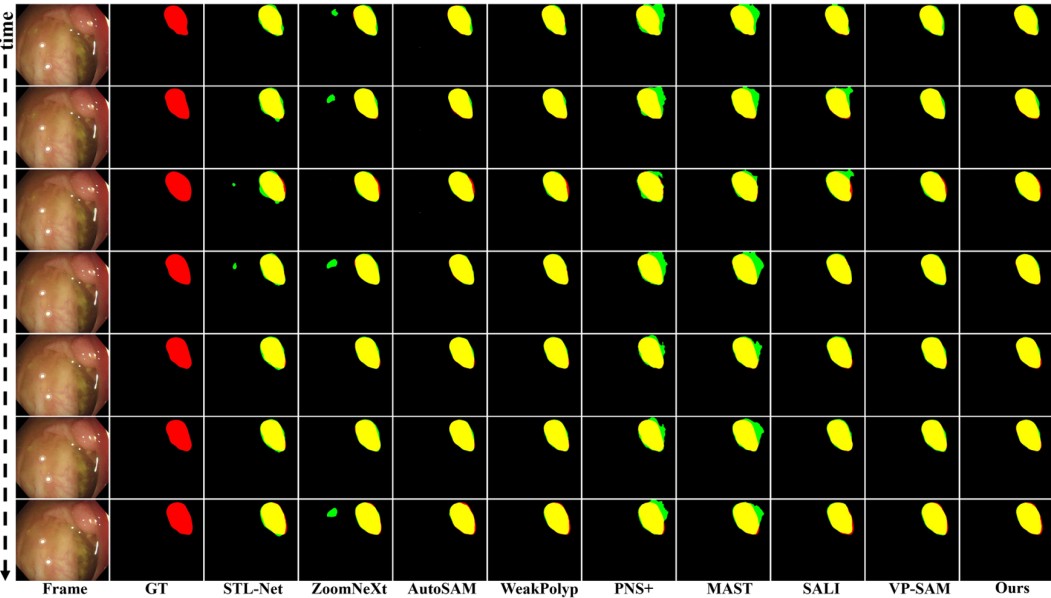

Figure 14: More frame-by-frame visual comparison of 7-frame video sequence on the SUN-SEG test set. (Red: GT, Green: Pred, Yellow: Overlap).

