# OpenReview forum: "WavePolyp: Video Polyp Segmentation via Hierarchical Wavelet-Based Feature Aggregation and Inter-Frame Divergence Perception"
_ICLR.cc/2026/Conference — ICLR 2026 Poster_

### Official Review · Reviewer_bCRF · 2025-10-30

**Soundness:** 2
**Presentation:** 2
**Contribution:** 2
**Rating:** 4
**Confidence:** 5

**Summary:**

This paper proposes WavePolyp, a novel framework for video polyp segmentation (VPS) that addresses two key challenges: intra-frame feature discrimination in highly camouflaged polyps and inter-frame temporal inconsistency. The method introduces two core components: a Hierarchical Wavelet-based Feature Aggregation (HWFA) module that decomposes and enhances multi-level features in the frequency domain, and Inter-frame Divergence Perception (IDP) blocks that explicitly model temporal changes between frames. Extensive experiments on SUN-SEG and CVC-612 benchmarks show that WavePolyp outperforms existing state-of-the-art methods while maintaining near real-time inference speed.

**Strengths:**

1. Comprehensive Experiments and Ablations: The paper provides thorough ablation studies to validate the contribution of each component (HFC, LFC, AFA, IDA, IDD). Results show consistent improvements across multiple metrics and datasets, with particularly strong gains on the challenging SUN-SEG-Hard subset.
2. Efficiency-Aware Design: The model achieves a good balance between accuracy and inference speed (23.04 FPS), making it suitable for real-time clinical applications.

**Weaknesses:**

1. Limited Conceptual Justification for Frequency-Based Design:
While the HWFA module is motivated by the idea that polyps exhibit discriminative features in both high- and low-frequency domains, the paper does not provide strong empirical or theoretical evidence for this claim. For example, visualizations of frequency-domain feature maps or analyses of polyp characteristics in the frequency space are missing.
2. Limited Conceptual Justification for Frequency-Based Design:
While the HWFA module is motivated by the idea that polyps exhibit discriminative features in both high- and low-frequency domains, the paper does not provide strong empirical or theoretical evidence for this claim. For example, visualizations of frequency-domain feature maps or analyses of polyp characteristics in the frequency space are missing.
3. Narrow Evaluation Scope: All experiments are limited to polyp segmentation. While the method is claimed to be general, no experiments on other video object segmentation benchmarks (e.g., DAVIS, YouTube-VOS) are provided to demonstrate broader applicability. The comparison with non-medical VPS methods is limited, making it difficult to assess the generalizability of the proposed components.
4. Lack of Novelty in Wavelet-Based Design and Inadequate Related Work:
The related work in Section 2.2 fails to establish a clear and substantive distinction between the proposed HWFA module and prior wavelet-based methods in vision tasks (e.g., SDWNet, LAAT, FEDER). The claim of novelty appears overstated. The application of DWT for feature decomposition, followed by attention mechanisms and hierarchical aggregation, is a known paradigm. The paper does not convincingly answer: What is the fundamental architectural or methodological leap here? The HWFA module, while well-engineered, comes across as a straightforward application of existing frequency-domain concepts to a new dataset, rather than a conceptual breakthrough.
5. The "Divergence Perception" in IDP is Unfocused and Potentially Noisy. The core mechanism of the IDP block is to compute a frame difference Diff = Shift(V) - V and modulate it via attention. However, this is a generic, low-level motion signal that captures all changes between frames, not just those pertaining to the polyp. This design is highly susceptible to noise from irrelevant background motion, such as camera jitter, fluid flow, or intestinal wall movements. The paper provides no evidence that the IDP block selectively focuses on polyp-specific divergence. Without an explicit mechanism to ground the divergence signal to the polyp region (e.g., using the current prediction as a guide), it is likely that the model is learning from a noisy signal, which could harm robustness in complex scenes. The failure cases in Fig. 7 (e-f) with overlapping intestinal walls might be a direct consequence of this issue.
6. The Overall Architecture Lacks Conceptual Novelty. The combination of DWT for multi-scale frequency analysis and attention mechanisms for feature refinement is a well-established design pattern in contemporary literature. The paper does not demonstrate that the proposed HWFA or IDP blocks represent a significant departure from this pattern. The HWFA module can be largely viewed as a specific instantiation of a multi-scale feature refinement network that uses DWT as its decomposition tool, rather than a fundamentally new operator. The incremental nature of the architectural design is a significant drawback for a paper aiming for a top-tier conference.
7.  Performance Gains Are Marginal and Not Compelling. A close inspection of Table 1 and Table 3 reveals that the performance advantage of WavePolyp is not decisive. On SUN-SEG-Hard, WavePolyp (Dice: 87.55) outperforms ZoomNeXt (Dice: 85.22) by 2.33 points, which is a solid but not groundbreaking improvement. More critically, on SUN-SEG-Easy, the gap is much smaller (WavePolyp: 88.96 vs. ZoomNeXt: 87.55, a 1.41-point difference).  When considering efficiency in Table 3, WavePolyp (23.04 FPS, 86.63M Params) and ZoomNeXt (22.48 FPS, 84.78M Params) are nearly identical in speed and model size, yet the performance delta is minimal. This suggests that the proposed complex modules (HWFA + IDP) offer only a marginal benefit over a much simpler and more general baseline, which severely undermines the claim of a significant advancement.

**Questions:**

See the weaknesses.

---

> ### Author Response · Authors · 2025-11-26
> **Response to reviewer bCRF**
>
> We thank the reviewer for the comprehensive review and for acknowledging our extensive ablation studies and efficiency-aware design. We value the constructive criticism regarding theoretical justification and novelty. In our revised version, we have further strengthened the theoretical justification, clarified our distinct contributions, and enriched the experimental validation to ensure the completeness of our study.
>
> **Q1&Q2: Limited conceptual justification for frequency-based design.**
>
> We agree that visualizing the internal representations is the most direct way to justify our design rationale.
> To further validate the effectiveness of HWFA, we visualize the final aggregated frequency features ($f_1^{HF}$ and $f_1^{LF}$) in Section 4.2(see Figure 5). These features represent the culmination of our hierarchical aggregation and are strategically fused into the network to refine specific scales.
> As shown in the visualization (Figure5, Col 3), this map sharply activates along the polyp boundaries and surface textures. This confirms that the HFC module successfully extracts high-frequency details, which are then integrated into the shallow layer ($x_1$) to recover fine-grained spatial details lost in the encoder.
> In contrast, the $f_1^{HF}$ (Figure5, Col 4) highlights the holistic lesion while suppressing background noise. This feature is integrated into the deep layer ($x_3$) to reinforce global semantic localization.
> Such kind of clear visual separation empirically proves that HWFA successfully disentangles discriminative cues (Texture/Boundary vs. Holistic Body) in the frequency domain (empirically verify Lines 279-284), justifying our "Decompose-Enhance-Aggregate" design.
>
> **Q3: Narrow evaluation scope (Lack of DAVIS/YouTube-VOS).**
>
> We appreciate the reviewer’s suggestion to assess the model’s generalizability on broader benchmarks. While DAVIS and YouTube-VOS are regarded as gold standards for general Video Object Segmentation (VOS), our proposed framework mainly focused on Video Polyp Segmentation (VPS) for the following reasons.
> General VOS targets distinct, often salient foreground objects. In contrast, VPS deals with highly camouflaged lesions that share nearly identical visual appearances (color/texture) with the background mucosa. Our proposed HWFA module is tailored to detect these subtle high-frequency and low-frequency anomalies.
> The temporal challenges in colonoscopy (e.g., fast peristalsis, fluid interference) differ fundamentally from natural video motion. Validating on large-scale polyp datasets ensures that the proposed model can effectively handle these clinical-specific temporal inconsistencies.
> Following established SOTA works in the VPS domain (e.g., PNS+, VP-SAM, SALI), we prioritized the authoritative SUN-SEG and CVC-612 benchmarks to demonstrate our contribution to this specific clinical problem. Certainly, should any general VOS datasets present challenges comparable to those in polyp videos (e.g., high camouflage and non-rigid deformation), we are confident that our method would effectively address these problems.
>
> **Q5: Concern that IDP uses noisy, unfocused divergence.**
> The IDP architecture is explicitly designed to mitigate noise through a semantic gating mechanism.
> We do not propagate the raw difference Diff directly. As shown in Eq. (11), the divergence signal is modulated by an Attention Map ($Att$) derived from the query ($Q$) and key ($K$) features. Crucially, since $Q$ and $K$ are extracted from the deep encoder/decoder layers, they contain high-level semantic information about the polyp’s location and appearance, rather than low-level pixel intensities.
> Unlike pixel-wise dense matching (e.g., Optical Flow) which is computationally heavy and sensitive to local noise, our attention mechanism operates along the temporal dimension, generating a $T \times T$ weight matrix.
> This design ensures negligible computational overhead.
> More importantly, it acts as a global semantic filter, assigning weights based on frame-level relevance. It learns to "gate" the divergence signal, suppressing irrelevant background motion (e.g., peristalsis) while focusing on the polyp's semantic consistency thereby effectively mitigating the temporal inconsistencies arising from significant inter-frame divergence and non-rigid polyp deformations.
> The effectiveness of this filtering is empirically supported by our new frame-wise tracking visualization in Appendix (see Figure13 and Figure14). The visualization demonstrates that while the baseline (w/o IDP) suffers from mask flickering due to background changes, the full model maintains consistent and stable tracking of the polyp across frames, confirming that the IDP block successfully filters out unfocused noise.

---

> ### Author Response · Authors · 2025-11-26
> **Response to reviewer bCRF**
>
> **Q4 & Q6: Novelty issues.**
>
> While DWT is a classic tool, our methodological innovation lies in the "Decompose-Enhance-Aggregate" paradigm, which fundamentally differs from the cited works.
> SDWNet and LAAT utilize wavelet transforms for image restoration to recover pixel details. In contrast, WavePolyp focuses on discriminative segmentation, aiming to extract semantic boundaries of camouflaged polyps.
> FEDER addresses foreground-background similarity by employing learnable wavelets, which introduces additional optimization complexity to find optimal frequency bands. In contrast, WavePolyp utilizes standard Discrete Wavelet Transform (DWT). We prove that complex learnable decomposition is not strictly necessary; instead, standard DWT is sufficient and efficient when paired with our novel enhancement and aggregation modules.
> We argue that high and low frequencies require distinct processing strategies, a perspective often overlooked.
> High-frequency components contain both textural details and noise. The HFC unit employs a residual attention mechanism to selectively amplify discriminative textural cues (polyp surface) while suppressing irrelevant noise, rather than processing all high-frequency signals equally.
> Since low-frequency components contain global semantic and illumination information, we introduce Position Normalization (PN) and Inverse PN in the LFC unit. This standardizes the statistical distribution of structural features, enhancing the model's robustness to lighting variations inherent in colonoscopy.
> Unlike standard concatenation, AFA employs a Window-based Linear Modulation mechanism. It calculates local affine coefficients to modulate the sharpened high-frequency features under the guidance of deep semantic features. This explicitly bridges the semantic gap, ensuring that the enhanced details are correctly aligned with the object's semantic location.
> We emphasize that applying our Hierarchical Wavelet-based Feature Aggregation (HWFA)—which leverages frequency decomposition to disentangle discriminative cues—is a novel and intrinsically suitable solution for VPS, effectively resolving the critical challenge of high lesion camouflage.
>
> **Q7: Performance gains are marginal and not compelling.**
>
> We argue that the performance gains of our model are statistically significant and clinically critical, especially when analyzing the detailed breakdown (Seen vs. Unseen) provided in Appendix A.4 (Table 7).
> The bottleneck of clinical diagnosis lies in "Hard" samples (small, camouflaged polyps). Our model consistently dominates in these difficult scenarios:
> We achieve 90.31% Dice vs. ZoomNeXt's 88.09%, a substantial gap of +2.22% on the SUN-SEG-Hard-Seen.
> We achieve 84.79% Dice vs. ZoomNeXt's 82.35%, a gap of +2.44% on the SUN-SEG-Hard-Unseen.
> This consistent >2.2% improvement proves that WavePolyp effectively tackles the core challenges of camouflage and motion, regardless of whether the patient data was seen during training.
> Beyond hard cases, our model shows superior generalization. On SUN-SEG-Easy-Unseen data, where baseline performance often drops, we outperform ZoomNeXt by +2.17% (84.87% vs. 82.70%). This confirms that our frequency-based features are robust and transferable.
> The smaller gap on Easy-Seen (+0.65%) is expected as foundational segmentation network already achieve satisfying performance (>92.40%). However, in medical AI, pushing the boundary on the remaining failure cases (Hard/Unseen) is far more valuable than incrementally improving already solved cases.
> Achieving these robust gains (especially on Hard/Unseen sets) while maintaining real-time speed (23.04FPS) demonstrates a superior efficiency-accuracy trade-off compared to the baselines.

---

### Official Review · Reviewer_YTma · 2025-11-01

**Soundness:** 3
**Presentation:** 3
**Contribution:** 2
**Rating:** 4
**Confidence:** 4

**Summary:**

This paper proposes WavePolyp, a method for video polyp segmentation (VPS) in colonoscopy videos. The model targets two key challenges: (i) high camouflage of polyps within surrounding tissue and (ii) significant inter-frame divergence (shape, size, position, boundary changes) that causes temporal inconsistency. WavePolyp contains two main components:

•	Hierarchical Wavelet-based Feature Aggregation (HWFA): Applies discrete wavelet transform (DWT) to multi-level encoder features, splits into low-frequency (LF) and high-frequency (HF) components, refines them via bespoke LF/HF calculation units (LFC/HFC) with attention and normalization strategies, and aggregates them across scales by an Ascending Frequency-guided Aggregation (AFA) unit. The goal is to excavate fine-grained intra-frame discriminative cues (edges, textures in HF; illumination/color in LF).

•	Inter-frame Divergence Perception (IDP): A temporal module that computes frame-wise differences via a temporal shift (TSM-like) operation, modulates them with a learnable projection, and applies a time-only attention to emphasize regions with meaningful temporal changes. A two-layer temporal convolution diffuses the divergence signal, and features are fused in a coarse-to-fine decoder.
Experiments on SUN-SEG (Easy/Hard splits) and CVC-612 show improvements over a set of baselines, including SLT-Net, ZoomNeXt, SALI, VP-SAM, and others, with competitive FPS on RTX 3090. Ablations attribute gains to both HWFA and IDP, and to specific internal design choices (normalization in LFC, AFA, divergent attention/diffusion). The paper reports favorable Dice, S-measure, E-measure, and weighted F across datasets, and discusses failure cases.

**Strengths:**

Originality:
•	Using DWT to explicitly separate HF/LF feature bands and tailor distinct processing (HFC/LFC) in a VPS context is a thoughtful design. While wavelets have been used in other CV tasks, positioning them to address polyp camouflage with a hierarchical aggregation (AFA) is a creative application.

•	The IDP block frames temporal modeling as learning inter-frame divergence via a simple yet principled TSM-based difference, time-only attention, and diffusion. It departs from heavier 3D/transformer-temporal designs by focusing on divergence emphasis with relatively low complexity.

•	The multi-scale zooming scheme via ZoomNeXt-style feature merging complements the frequency-domain design, potentially improving robustness to size variation.

Quality:
•	Comprehensive experimental protocol on standard VPS datasets (SUN-SEG with Easy/Hard splits; CVC-612), multiple metrics, and a reasonably broad set of recent baselines, including both image and video methods.

•	Ablation studies that dissect both macro-components (HWFA vs. IDP) and micro-components (HFC/LFC normalization, AFA, IDP’s attention/diffusion), plus exploration of clip length.

•	Efficiency analysis (params, GFLOPs, FPS) against video methods provides a useful view of the accuracy-efficiency trade-off.

Clarity:
•	The high-level motivation and module roles (camouflage handled by HWFA; temporal divergence by IDP) are clearly articulated and connected to the challenges illustrated in Fig. 1.

•	The pipeline overview (encoder with zoomed scales, HWFA, IDP decoder, final predictor) is easy to follow.

•	Equations and data flow for IDP are described sufficiently to reproduce the idea. The HFC/LFC and AFA figures are helpful.

Significance:
•	VPS is a clinically meaningful and actively researched area; improvements that bring both accuracy and near-real-time speed are valuable.

•	The approach offers a perspective shift: deliberately leveraging frequency-domain cues for medical temporal segmentation, which could transfer to other endoscopic or camouflaged lesion videos.

**Weaknesses:**

Novelty vs. prior art:
•	Wavelet/decomposition-driven feature processing has been employed in medical segmentation (e.g., FEDER and related frequency-decomposition works) and image generation/denoising; the paper cites some but does not thoroughly differentiate its technical novelty beyond the specific arrangement (HFC/LFC/AFA) and application to VPS. More rigorous positioning against frequency-aware segmentation works (e.g., learnable wavelet bases, frequency-aware attention, Fourier-based filtering such as FFC/AFNO/FFT-based U-Nets) is needed to substantiate novelty.

•	The IDP block conceptually resembles several recent “temporal difference + attention” modules: TSM/TDN-like frame differencing, TEA-style excitation of motion, and time-attention in transformer decoders. The paper cites TSM but not closely related temporal-difference attention models. A tighter comparative discussion is necessary to establish conceptual distinctiveness.


Methodological specification:
•	The AFA unit is central, but the mathematical description is incomplete. The appendix starts an optimization objective for LF AFA but cuts off; the main text mentions a “window-based linear model” inspired by Swin Transformer, yet does not concretely define the window filtering/weighting, parameterization, loss, or computations. Reproducibility of AFA as described is difficult.

•	The LFC normalization stack mixes PN/IPN with BN in Eq. (2), although the text claims BN in HFC and IN in LFC; there is an inconsistency: the formula shows BN in LFC after PN, contrary to the stated IN replacement. If IN is truly used, the equation should reflect it. Moreover, the learnable µ′, σ′ details (how they are produced per position, per channel? with which conv kernel sizes? affine parameters?) are underspecified.

•	For IDP, the shapes and exact axes for attention are a bit ambiguous: K, Q in R^{HW × T} suggests flattening spatial into tokens and attending over time only, but the normalization over sqrt(HW) is unusual (typically sqrt(d)). If d=HW, this is extremely large and may cause scale issues; justification or reparameterization (e.g., linear projections to lower d) is not provided. Complexity claims should be reconciled with potentially large HW × T multiplications.

•	The learnable WV is a T × T matrix. If T is variable at test time, how is WV handled? If fixed T=5 always, this should be stated as a constraint. Also, the two T × 3 × 3 convolutions imply grouping or depthwise temporal convolution per frame with spatial 3x3 kernels, but the notation is nonstandard; more detail would help.

Experimental rigor:
•	The baseline list is decent, but misses some recent strong VPS/medical video temporal models (e.g., DINOv2-temporal adaptations, token mixers with long-range memory, or optical-flow-based medical segmentation), and does not compare to simple but strong baselines like “per-frame SOTA IPS + temporal smoothing/post-processing.” Including a naive temporal regularization baseline could calibrate the added value of IDP.

•	Statistical significance or variability is not reported (no std/CI across runs or seeds). Given close margins over VP-SAM and SALI on some metrics, significance analysis is warranted.

•	The clip length ablation is limited to a single dataset and reports only a curve; numeric values and an analysis of inference-time latency/memory vs. T would be useful for practitioners.

•	The speed benchmark uses batch size 1 on RTX 3090; ensuring comparable precision settings (FP32 vs. AMP), input size, and consistent pre- and post-processing is important. Also, the model depends on PVTv2-B5 and ZoomNeXt merging; an ablation on the backbone and the multi-scale merging would contextualize where the gains originate.


Generalization and data:
•	Only two datasets (one large, one small) are used; cross-dataset generalization (train on SUN-SEG, test on CVC-612 without fine-tuning) would be informative.

•	Robustness to common endoscopic artifacts (specular highlights, motion blur, occlusions), lighting changes, and strong peristalsis is only qualitatively discussed; quantitative robustness tests (e.g., controlled corruption benchmarks) are missing.

•	The method benefits from training on three zoom scales; at inference, is the multi-scale encoder always used (computational overhead)? The text implies that multi-scale features are merged at 1.0x; an explicit statement of the inference-time pipeline and cost would help.


Clarity/consistency issues:
•	Some figures/equations appear truncated or inconsistent (e.g., AFA math in appendix). Equation (8) references DWTh/DWTl applied to fk; it’s not entirely clear whether DWT is applied at each level to encoder features vs. features already processed by prior steps. A tidy pseudo-code or module-by-module dimensionality flow would resolve ambiguities.

•	The reliance on ZoomNeXt’s multi-scale merging is nontrivial; a short recap or an ablation “w/o multi-scale merging” is needed to attribute gains correctly to HWFA/IDP vs. upstream feature enrichment.

**Questions:**

1.	AFA specification and reproducibility:
•	Please provide the complete mathematical definition of AFA, including the “window-based linear model” parameters, the guidance weighting using W^HF/LF, and how the downsampled aggregated feature f_k^dh is computed. Is it a learned linear filter per window guided by W? Any regularization?

•	In Appendix A.1, the optimization objective for LF AFA is truncated. Can you provide the full objective and solution or implementation details?

2.	Normalization details in LFC:
•	The text says BN in HFC and IN in LFC, but Eq. (2) shows BN in LFC. Which is correct? If IN is used, please provide the exact formula and where PN/IPN sit relative to IN. How are µ′, σ′ computed (per-position, per-channel), and with what kernel sizes/parameterization?

3.	IDP attention scaling and complexity:
•	You use Softmax(K^T Q / sqrt(HW)). Why is sqrt(HW) the appropriate scale? Do you use linear projections to reduce dimensionality before attention? If not, what are the memory/time costs for HW × T tokens, especially with 352×352 inputs? Please include exact tensor shapes and a complexity analysis.

•	WV is T × T and appears to hardcode the clip length. How does the module adapt to different T at test time, or is T fixed? Would a content-dependent dynamic filtering (e.g., 1D temporal conv) be preferable?

4.	Baseline completeness and significance:
•	Could you include comparisons with: (i) per-frame SOTA polyp segmenter plus temporal smoothing (e.g., Savitzky–Golay, EMA), (ii) an optical-flow-guided refinement baseline, and (iii) a temporal transformer with time-only attention but without divergence computation? This would isolate the incremental value of your divergence design.

•	Please report mean±std across at least 3 runs for key metrics to ascertain the significance of improvements, especially over VP-SAM and SALI.

5.	Multi-scale pipeline at inference:
•	Do you run the tri-scale encoder (0.75×/1.0×/1.25×) at inference? If yes, what is the added latency and memory compared to single-scale? If not, how is multi-scale training transferred to single-scale inference?

6.	Wavelet choices and learned variants:
•	Which wavelet family (e.g., Haar, Daubechies) is used? Have you explored learnable wavelet filters or alternative frequency decompositions (e.g., DCT/Fourier), and how do results compare?

•	How sensitive is performance to DWT level, number of scales, and the balance between HF/LF weighting?

7.	Generalization and robustness:
•	Please include cross-dataset generalization results (e.g., train on SUN-SEG, test on CVC-612 without fine-tuning) and robustness to common perturbations (blur, noise, brightness shifts), to validate claims of robustness to camouflage and inter-frame divergence.

8.	Clinical practicality:
•	VP-SAM requires a point prompt; your method does not. However, your reliance on multi-scale processing and wavelet decomposition adds overhead. Can you provide a detailed breakdown of latency per module and avenues for lightweight deployment (e.g., replacing PVTv2-B5, pruning, quantization), including any preliminary results?

---

> ### Author Response · Authors · 2025-11-26
> **Response to reviewer YTma**
>
> We sincerely thank the reviewer for the meticulous and comprehensive review, and for recognizing the originality of our wavelet-based design and the rigor of our experimental protocol. In this rebuttal, we would like to make efforts to address your concerns through the real-time feedback.
>
> **Q1: AFA specification, reproducibility, and mathematical definition.**
>
> The AFA module is mathematically grounded in the differentiable weighted fast guided filter and swin transformer. Please refer to Eq. (3-6) and Lines 256-258.
> The guidance weight maps ($W^{HF}/W^{LF}$) are derived directly from the HFC and LFC units (specifically via the CSA attention mechanism, Eq. 1). They serve as spatial attention maps to prioritize discriminative regions during the optimization process.
> The "window-based linear model" parameters ($\sigma_w, \mu_w$) are dynamically computed for each window by solving the weighted least squares optimization in Eq. (4). They are not fixed learnable kernels, but closed-form solutions derived from local statistics (mean/variance). This content-adaptive operation allows the module to precisely align and fuse discriminative details based on the specific texture and structure of each image.
> As defined in Eq. (3) and Eq. (5), $f_k^{dh}$ represents the downsampled aggregated feature map resulting from the linear projection. This design follows the fast guided filter paradigm to significantly reduce computational complexity while preserving edges.
> For regularization, it is strictly applied. As explicitly shown in Eq. (4), the term $\epsilon \sigma_w^2$ serves as Ridge Regularization to ensure numerical stability.
> We respectfully clarify that the optimization objective for LF AFA in Appendix A.1 is not truncated. It is structurally identical to the HF formulation (Eq. 4), with the only distinction being the input features ($f^{LF}$ instead of $f^{HF}$). The provided equation represents the complete ridge regression objective.
> Regarding reproducibility, as explicitly stated in the Abstract, we will release the complete source code upon publication.
>
> **Q2: Inconsistency regarding BN vs. IN in LFC and Eq. (2).**
>
> We apologize for the clerical error in Eq. (2). As correctly described in the text (Section 3.2), the normalization layer used in LFC is indeed Instance Normalization (IN), not Batch Normalization (BN). This choice is deliberate to handle the global statistics of low-frequency components.
> We have corrected Eq. (2) in the revised manuscript.
> The affine parameters ($\mu', \sigma'$) are computed in a per-position manner (calculating mean/std along the channel dimension, preserving spatial structure). To capture local structural dependencies, $\mu$ and $\sigma$ are updated via a lightweight sub-network consisting of three convolutional layers, including Layer 1: $1\times1$ Conv ($C_{in}=1, C_{out}=16$); Layer 2: $3\times3$ Conv ($C_{in}=16, C_{out}=16$); Layer 3: $1\times1$ Conv ($C_{in}=16, C_{out}=1$). This design allows the network to dynamically generate spatially-adaptive affine parameters for effective feature restoration.
>
> **Q4: Baseline completeness (Temporal Smoothing, Optical Flow, Time-only Attention).**
>
> We appreciate the reviewer's suggestions to isolate the value of IDP. However, based on the unique characteristics of endoscopic data and prior literature, we respectfully clarify why these specific baselines were not prioritized:
> While standard in natural videos, optical flow is widely considered unreliable for colonoscopy. As analyzed in SALI, the low texture contrast between polyps and the mucosal background, combined with fluid motion and reflections, leads to noisy flow estimation. Introducing optical flow often introduces more error than valid motion cues in this specific domain.
> We have effectively compared against methods using time-only attention. Methods like PNS+ and MAST (included in our Table 1) rely heavily on generic temporal self-attention mechanisms without explicit divergence modeling.
> Temporal smoothing is ill-suited for VPS due to the specific challenges of colonoscopy videos---polyps undergo rapid, irregular morphological changes due to intestinal peristalsis.
> Besides, probe’s movements and jitters may cause polyps’ dramatic position shift and size variations between frames.
> Applying rigid smoothing algorithms (like EMA) would act as a low-pass filter, blurring the boundaries of fast-deforming polyps or causing "ghosting" artifacts during rapid movement, ultimately degrading the segmentation of hard samples.

---

> ### Author Response · Authors · 2025-11-26
> **Response to reviewer YTma**
>
> **Q3: Justification for $\sqrt{HW}$ scaling, dimensionality reduction, and handling variable $T$.**
>
> We thank the reviewer for the detailed technical scrutiny. We clarify the exact tensor shapes and operations below:
> There is a misunderstanding regarding the input resolution. The IDP module does not operate on the raw $352 \times 352$ image.
> The inputs to the decoder ($x_1$ to $x_4$) are the extracted features from the HWFA module. Their channel dimension is unified to $C=64$, and the spatial resolutions range from $88 \times 88$ (for $x_1$) down to $11 \times 11$ (for $x_4$).
> Consequently, the memory and time costs are calculated on these compact feature maps, not on the high-resolution input. The attention complexity $\mathcal{O}(T \cdot C^2 \cdot HW)$ is negligible given that $H, W$ and $T$ (clip length=5) are small. Empirically, under this setting, the inclusion of the IDP module consumes only ~30MB of memory and adds 8.8ms of latency, validating its high efficiency.
> The scaling factor $\sqrt{HW}$ is based on the variance-preserving principle. In our IDP, the dot product aggregation is performed over the spatial dimensions ($H \times W$) to compute global temporal affinity.
> Since the summation involves $H \times W$ elements (e.g., $11 \times 11$ to $88 \times 88$), the variance of the logits scales linearly with the spatial area.
> Dividing by $\sqrt{HW}$ normalizes this variance, preventing the Softmax gradients from vanishing, which is standard practice for attention mechanisms aggregating over a variable number of tokens.
> The clip length $T$ is a structural hyperparameter (ablated in Figure 6 of our paper) that determines the model's temporal receptive field.
> Therefore the matrix $W_V$ ($T \times T$) corresponds to the fixed temporal window size (e.g., $T=5$). To handle video sequences of arbitrary length, we adopt the standard Sliding Window strategy during both training and inference. This ensures that the model processes a consistent temporal context matching the learned weights.
>
> **Q5: Multi-scale pipeline at inference (Latency & Strategy).**
>
> The tri-scale encoder (0.75×, 1.0×, 1.25×) is utilized during inference, maintaining consistency with the training phase.
> Compared to the single-scale baseline (Latency: 24.4ms, Memory: 1.11GB), the tri-scale pipeline adds 18.8ms in latency (Total: 43.2ms) and increases memory usage by 1.78GB (Total: 2.89 GB). While the cost increases, it brings a significant performance gain of +0.6% in Dice on SUN-SEG-Hard test set, which we believe is a worthy trade-off for high-precision medical scenarios.
> Alternatively, if efficiency is the primary constraint, the zooming strategy can be omitted. Even in this single-scale configuration, our method retains its SOTA superiority over existing baselines.
>
> **Q6: Wavelet family used; exploration of learnable filters or alternatives (DCT/Fourier).**
>
> We utilize the standard Haar wavelet for the Discrete Wavelet Transform (DWT). We selected Haar for its computational simplicity and effectiveness in preserving edge information without introducing boundary artifacts.
> We deliberately chose standard DWT over learnable wavelets (e.g., FEDER) or Fourier transforms (DCT/FFT) based on two core design principles:
> As emphasized in Section 3.2, a key advantage of our HWFA is being parameter-free in the decomposition stage. While learnable filters offer flexibility, they increase model complexity and optimization difficulty. We demonstrate that fixed DWT is sufficient when paired with our strong aggregation module (AFA).
> Unlike Fourier transforms (DCT/FFT) which capture global frequency responses but lose local spatial information, DWT inherently preserves spatial structure while decomposing frequencies. This property is critical for segmentation tasks where precise boundary localization is paramount.
> We only apply 1-level DWT at each stage. We have extensively analyzed the impact of feature scales in Appendix A.2 (Table 4). We compared fusing features starting from different encoder stages ($f_1$ to $f_4$).
>
> **Q7: Cross-dataset generalization and robustness analysis.**
>
> We appreciate the reviewer's valuable suggestions to validate the reliability of our method.
> We have included the cross-dataset generalization results (Training on SUN-SEG, Testing on CVC-612) in Appendix A.4 of the revised manuscript.
> To further demonstrate robustness against camouflage and inter-frame divergence, we have provided additional visual ablations for both HWFA and IDP, as well as frame-by-frame visual comparisons on continuous video clips in Appendix A.5 (see Figure13 and Figure14). These results consistently confirm the stability and generalization capability of WavePolyp in challenging scenarios.

---

> ### Author Response · Authors · 2025-11-26
> **Response to reviewer YTma**
>
> **Q8: Clinical practicality, latency breakdown, and lightweight deployment avenues.**
>
> Unlike VP-SAM which requires manual point prompts, WavePolyp is fully automated. This is critical for real-time clinical screening where human-in-the-loop interaction is often impractical.
> We performed a detailed resource analysis (on RTX 3090) to demonstrate deployment feasibility.
> The baseline(w/o HWFA and IDP) consumes 2589MB memory with 24.3ms latency.
> The IDP module is highly efficient, adding only +29MB memory and +8.8ms latency. The HWFA module adds +281MB memory and +10.8ms latency.
> The full model adds a total of ~300MB memory and ~19ms latency compared to the baseline. These specific overheads are well within the capacity of standard medical workstations, confirming that our core modules are computationally manageable.
> For resource-constrained devices, we explored switching the backbone from PVTv2-B5 to PVTv2-B2.
> his lightweight version drastically reduces memory usage to 1874.6MB (vs. 2890MB) and latency to 25.6ms (vs. 43.2ms), with only a marginal performance drop (0.7% Dice on SUN-SEG-Hard).
> This confirms that WavePolyp is highly robust and scalable, allowing for flexible deployment depending on hardware constraints.
> Even without any adjustments, we have already achieved 23.04 FPS, satisfying the standard real-time requirement for endoscopy.
> Further optimizations like pruning remain as future work.

---

### Official Review · Reviewer_1y5o · 2025-11-01

**Soundness:** 2
**Presentation:** 2
**Contribution:** 2
**Rating:** 4
**Confidence:** 4

**Summary:**

The paper presents a video polyp segmentation framework composed of two main modules: Hierarchical Wavelet-based Feature Aggregation (HWFA) and Inter-frame Divergence Perception (IDP) blocks. In the encoder, the HWFA module amplifies information from both high- and low-frequency features to address the camouflage property of polyps. The IDP module in the decoder captures inter-frame divergence to enhance temporal consistency.
Main contributions:
- The framework considers both intra-frame discriminative features and inter-frame divergence for improved segmentation performance.
- It accounts for both segmentation accuracy and real-time efficiency by evaluating GFLOPs and FPS.
- An extensive ablation study is conducted to validate the effectiveness of each component.

**Strengths:**

- Extensive experiments are conducted across multiple metrics, including GFLOPs and FPS, along with visualizations such as t-SNE plots for clearer interpretability.
- A thorough ablation study is performed, analyzing the impact of the presence or absence of different modules.
- Results are comprehensively compared against recent SoTA methods.

**Weaknesses:**

- The author mentions that they compared their results on two video polyp datasets, one of which is CVC-612. However, it is a static dataset. This dataset contains individual frames extracted from colonoscopy videos, which are static images and do not include temporal relationships. Therefore, validating a video model on such a static dataset may not effectively evaluate the video-specific contributions of the proposed model.
- The train-test split for CVC-612 is not described and should be specified.
- Since the IDP module is designed to capture inter-frame information, some frame-wise qualitative results are needed to validate its effectiveness.
- In Fig. 3, the CA+SA module is shown, but it is not discussed in detail in the text.
- In Table 2, it appears that when all modules except AFA are used, performance drops even below the baseline (top row). This behavior requires justification.

Minor:
In Section 3.1, the sentence “Due to the dynamic nature of video” should be rephrased, as it seems incomplete or unclear.

**Questions:**

- It should be explained how evaluating the model on CVC-612, a static dataset without temporal relationships, effectively reflects the video-specific contributions of the proposed framework.
- The train-test split for CVC-612 should be provided to clarify how the dataset was used for training and evaluation.
- Frame-wise qualitative results or visualizations should be provided to demonstrate the effectiveness of the IDP module in capturing inter-frame divergence.
- Description for the CA+SA module should be provided, as it is shown in Fig. 3.
- Justification should be provided for why performance drops below the baseline in Table 2 when all modules except AFA are used.

---

> ### Author Response · Authors · 2025-11-26
> **Response to reviewer 1y5o**
>
> We sincerely thank you for the affirmative and constructive comments. In this rebuttal, we would like to make efforts to address your concerns through the real-time feedback.
>
> **Q1: Concerns about CVC-612 being a "static" dataset without temporal relationships.**
>
> We respectfully clarify that CVC-612 (CVC-ClinicDB) is inherently a video-based dataset, preserves short-term temporal relationships between adjacent frames. As officially defined, the dataset encompasses 612 standard-definition frames from 31 video sequences of 23 patients. While the dataset often used for image segmentation, the frames within each sequence preserve their sequential order and temporal correlations. Compared to the large-scale SUN-SEG, validating on CVC-612 better evaluates a VPS method's capability to segment short-term clips. Our significant performance gain on CVC-612 (Table 1) specifically demonstrates the effectiveness of our IDP module in modeling these inter-frame temporal dependencies.
>
> **Q2: Train-test split for CVC-612.**
>
> The train-test split is explicitly described in Section 4.1 (Implementation Details).
> We state: "For CVC-612, we split the training set, validation set, and test set with a ratio of 6:2:2". This random split strategy at the clip level is a standard practice in VPS literature (e.g., An Embedding-Unleashing Video Polyp Segmentation Framework via Region Linking and Scale Alignment) to ensure fair comparison.
>
> **Q3: Request for frame-wise qualitative results to validate IDP.**
>
> We provide a frame-wise visualization ablation study in the Appendix (see Figure13 and Figure14).
> This visualization compares "Ours w/o IDP" vs. "Ours (Full components)" across continuous video sequences. Compared to the baseline without IDP, method equipped with IDP explicitly demonstrates the capacity for suppressing mask flickering and stabilizing tracking during rapid colonoscopy camera movement.
>
> **Q4: Details on the CA+SA module in Fig. 3.**
>
> The CA+SA module refers to the sequential application of Channel Attention (CA) and Spatial Attention (SA). Inspired by the standard CBAM design, we specifically tailored the pooling strategies to align with the distinct properties of frequency-domain features.
> In our HFC/LFC units, this mechanism helps the network selectively emphasize informative frequency components (e.g., texture in HF, boundaries in LF). Specifically, the CA module employs 1D convolution for efficient cross-channel interaction, dynamically selecting Average Pooling for Low-Frequency Components (LFC) to capture global background information. On the other hand, Max Pooling is applied for High-Frequency Components (HFC) to highlight discriminative textures. The SA module utilizes a convolutional bottleneck structure to refine spatial regions.
> We have added the detailed mathematical formulation and description to the Appendix A.1 in the revised version.
>
> **Q5: Why does performance drop below baseline without AFA (Table 2)?**
>
> The configuration "w/o AFA" implies utilizing HFC/LFC to enhance features but fusing different feature levels (e.g., $f_{k}^{HF}$ and $f_{k+1}^{HF}$) via naive concatenation instead of the proposed AFA.
> Features at different scales possess inherent semantic gaps and spatial misalignments. The HFC/LFC units explicitly amplify specific frequency components (e.g., sharp edges/textures in HF).
> In the baseline, the network tolerates slight misalignments from naive fusion due to smoother feature maps. However, in "w/o AFA", simply forcing the strongly enhanced and sharpened features together without AFA's adaptive alignment (window-based linear modulation) results in amplified spatial artifacts and noise.
> These artifacts disrupt the decoder more severely than the unenhanced baseline features, validating that AFA is indispensable for aligning and aggregating these hierarchically enhanced frequency cues.
> Actually, we briefly noted this phenomenon in the original manuscript (stating 'Significant performance drops are observed when AFA is excluded'), but we acknowledge that the underlying theoretical cause was not sufficiently elaborated.
> We have revised Section 4.2 to explicitly explain this phenomenon, clarifying the necessity of AFA in bridging the semantic gap between frequency components.
>
> **Q6: Sentence in Section 3.1.**
>
> We have rephrased the sentence to improve clarity and grammatical correctness: "Due to the dynamic nature of video, characterized by temporal changes between frames, polyp segmentation often suffers from temporal inconsistencies."

---

### Meta-Review · Area_Chair_LZMd · 2026-01-07

**Summary:**

This paper presents a carefully engineered framework for video polyp segmentation, combining frequency-domain feature aggregation with explicit modeling of inter-frame divergence. Reviewers generally agree that the method is effective, well implemented, and supported by extensive experiments. The main concern across reviews is that the overall contribution is incremental. Reviewers YTma and bCRF note that the core ideas build on existing frequency-aware and temporal-difference designs. Nevertheless, the integration is coherent and well motivated for the target application, making this a borderline case.

**Reviewer Concerns:**

The rebuttal largely resolves concerns from Reviewer 1y5o regarding dataset usage, missing visualizations, and unclear module descriptions. Many technical and clarity issues raised by Reviewer YTma were also addressed, improving confidence in the implementation. Concerns from Reviewer bCRF about novelty and broader generalization are only partially addressed. Overall, the rebuttal strengthens the paper’s soundness and clarity, while some questions about conceptual novelty remain, supporting a borderline accept decision.

**Reviewer Scores:**

The rebuttal addresses the main concerns of Reviewer 1y5o regarding dataset usage, visual evidence, and module clarity. With these issues largely resolved, Reviewer 1y5o is likely to view the work more favorably and may increase the score. Meanwhile, Reviewer YTma is likely to increase the score modestly due to improved clarity and additional analyses. Although concerns about novelty persist, the added visualizations and expanded justification improve the empirical support of the method. Reviewer bCRF may slightly revise the score upward, though the assessment is likely to remain cautious.

---

### Decision · Program_Chairs · 2026-01-26

Accept (Poster)